# RAD3D-Prefix: Anomaly-Aware Prefix Learning on Frozen LLM for 3D CT Image to Report Generation

## Abstract

Recent advances in multimodal learning, including large language models (LLMs) and vision-language models (VLMs aka foundational models), have demonstrated strong adaptability to natural images. However, extending their use to the medical domain, particularly for volumetric (3D) images, is challenging due to high computational complexity and the need to model volumetric dependencies. The significant misalignment between visual and textual features further limits the ability to leverage the strength of LLMs, and naively fine-tuning these models on limited medical data often leads to overfitting and underperformance on downstream tasks. In this study, we address these challenges for volumetric radiology scans (specifically CT) report generation by introducing a simple, lightweight approach that minimizes the need for extensive parameter training. Our solution, called **RAD3D-Prefix**, employs a novel anomaly-aware prefix learning module that effectively aligns visual features from 3D images with textual features. This module integrates image embeddings with multi-label diagnostic classification logits, preserving critical clinical details while bridging the vision-language gap. By keeping the LLM frozen, our method requires minimal trainable parameters and mitigates the risk of overfitting on small, domain-specific datasets. Across four different evaluation criteria, **RAD3D-Prefix** outperforms existing similar-sized models and performs comparably to larger models that have more than five times the number of trainable parameters. Our approach demonstrates superior clinical relevance and out-of-domain generalization, highlighting the effectiveness of our lightweight, anomaly-aware prefix projection module. [1]

## 1 Introduction

Large Language Models (LLMs) are pre-trained on a massive amount of text, which allows them to generalize effectively and perform well on new tasks with no or few examples in downstream tasks involving zero-shot learning Sanh et al. (2021); Ramesh et al. (2021) and few-shot in-context learning Brown et al. (2020); Wei et al. (2022). These remarkable properties of LLMs have inspired their adoption in various vision-based tasks for multimodal applications Guo et al. (2023); Li et al. (2024). Most of these approaches concentrate on end-to-end training or fine-tuning using domain-specific image-text pairs Li et al. (2022); Kim et al. (2021). Specifically, multimodal models that work with medical images and text often rely on fine-tuning because the general data used to pre-train LLMs contains very few medical examples Lai et al. (2024). Specialized biomedical LLMs have been developed such as **BioGPT** Luo et al. (2022) and **BioMedLM** Bolton et al. (2024), to address the limitations of general LLMs in the medical field. These models are pre-trained on extensive biomedical text corpora like **PubMed** White (2020) to improve their ability to recognize medical terminology. Despite this specialized training, a significant challenge remains: the **modality gap** between medical images and textual data. This gap is particularly problematic for applications where accuracy is critical, such as generating clinical reports from 3D radiology images.

Generating clinical reports from radiology images requires aligning visual cues in the images with the domain-specific text in the reports. This is a challenging task because the 3D (volumetric) im-

---

[1]The source code and the pre-trained models will be made public after the review process.

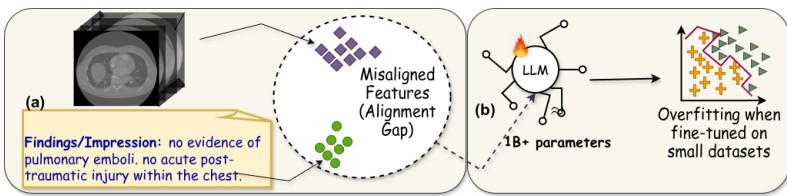

Figure 1: Two critical challenges involved in the report generation task: (a) Difficulty in achieving alignment between visual and textual semantics due to their inherently different characteristics, (b) Fine-tuning larger LLMs ($\approx 1B+$ parameters) on small domain-specific datasets leads to overfitting.

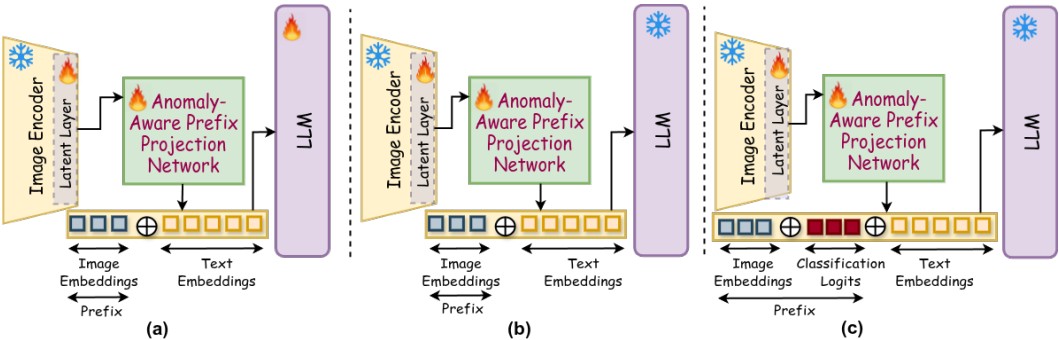

Figure 2: Three variations of the proposed projection module: (a) *V-1*: prefix includes image embeddings and involves LLM's fine-tuning (suitable for smaller LLMs), (b) *V-2*: prefix includes image embeddings with frozen LLM (suitable for larger LLMs), and (c) *V-3*: prefix includes image embeddings, and classification logits with frozen LLM.

ages contain comprehensive diagnostic information as compared to 2D images, and unlike regular captions, the clinical reports have long sequences and domain-specific language. Fine-tuning larger LLMs on limited medical image-text pairs often leads to overfitting. This process also requires optimizing a large number of trainable parameters, which creates significant computational overhead. To address these challenges (as illustrated in Fig 1), in this paper, we propose *RAD3D-Prefix* model with a ***lightweight anomaly-aware prefix projection*** module that allows faster training with minimal trainable parameters, while effectively aligning image and text features and retaining critical diagnostic cues. To this end, our proposed projection module generates a *prefix*, which is a fixed-length embedding sequence that represents both image features and multi-label diagnostic classes. This lightweight approach allows LLMs to be used in a zero-shot learning setting. Our method can even outperform domain-specialized LLMs with a similar parameter count.

To systematically demonstrate the impact of our proposed approach, inspired by Mokady et al. (2021); Wang et al. (2023), we investigated three different experiment setup (variant), as illustrated in Fig 2. In variant *V-1* (Fig 2 (a)), our projection network and LLMs are simultaneously trained with prefixes comprising of image embeddings alone, which are then concatenated with text embeddings. In variant *V-2* (Fig 2 (b)), the same prefix settings are used with frozen LLM. In variant *V-3* (Fig 2 (c)), we incorporate the diagnostic details by concatenating multi-abnormality classification logits with image prefix representation as projection input for frozen LLM. We examined these approaches using smaller models (with a few million parameters) and larger models ($\approx 1B+$ parameters). Unlike Wang et al. (2023), our work focuses on improving the core processing of 3D image embeddings and their projection as input to LLMs while preserving clinically significant multi-abnormality entity markers. Our experiments show that larger LLMs ($\approx 1B+$ parameters) demonstrate strong zero-shot or few-shot capabilities when frozen, as compared to fine-tuned version. This suggests that overfitting hinders their inherent knowledge and representational capacity when trained on small domain-specific datasets. In contrast, smaller models (with a few million parameters) benefit from fine-tuning, as their limited parameter count restricts their generalization ability without adaptation. Our main contributions are summarized below:

- **Novel lightweight anomaly-aware prefix projection module:** We propose a novel *lightweight anomaly-aware prefix projection module* to generate clinical reports for 3D radiology images with minimal parameter training. In contrast to existing 2D image-based vision-language models (VLMs) Moor et al. (2023), **RAD3D-Prefix** is the first to align 3D image embeddings and anomaly logits with a frozen LLM. Thus, narrowing the vision-language gap, unlike natural image models Dai et al. (2023); Jin et al. (2024), which face semantic gaps, especially when deployed in the medical domain. While the basic prefix learning concept exists Mokady et al. (2021), our anomaly-aware 3D medical imaging approach with visual and classification cues integration is novel and clinically impactful.

- **Extensive experiments on the influence of prefix design and LLM tuning strategies:** We conducted extensive experiments on three different model variations to determine the most effective approach for different prefix designs. Further, we performed comparison across *five* LLMs from 96.1M to 1.6B parameters, with frozen vs fine-tuned setups, where pretraining data are entirely non-medical. This provides actionable guidance (fine-tune $<$ 1B, freeze $\approx$1B+) that has not been studied in 3D medical imaging and perform differently compared to natural images (LLaVA Liu et al. (2023) and BLIP-2 Li et al. (2023b)).

- **Ensure clinically relevant outputs:** We incorporated multi-anomaly classification logits to retain important clinical details in the generated reports. This explicitly exposes clinical concepts (e.g., effusion, consolidation) to the LLM. Additionally, we used medical-specific metrics for validation to ensure diagnostic precision.

- **Outperforms similar-sized and domain-specialized models, while performing comparable with larger models:** Our proposed method, despite minimal training, empirically outperforms existing techniques when using frozen LLMs with the same parameter count and specialized domain pre-training. The model also performed comparably to methods employing frozen LLMs with higher parameter count, supported by bootstrap analysis for statistical significance. Furthermore, using the same vision encoder across all methods shows that the gains stem from the anomaly-aware prefix rather than a heavier backbone.

## 2 RELATED WORK

### 2.1 MEDICAL REPORT GENERATION

Llava-Med Li et al. (2023a), Med-Flamingo Moor et al. (2023), and Med-PaLM Singhal et al. (2023) are the major models designed for medical report generation that use Vision-Language Models (VLMs) trained on extensive image-text datasets. However, these models have a key limitation: they can't process 3D medical images like CT and MRI scans because of the high complexity and computational costs involved. To address this, other models like, CT2Rep Hamamci et al. (2024b), CT-AGRG Di Piazza et al. (2024), E3D-GPT Lai et al. (2024), and Med-2E3 Shi et al. (2024) have been developed. These solutions capture global features from 3D images and use them as input for text decoders to generate reports.

The **CT2Rep** model uses a 3D medical vision encoder to extract global features from CT images and integrate them into a language model for report generation, showing initial effectivness. Building on this, the **CT-AGRG** model incorporates abnormality-guided recognition, which allows the framework to detect anomalies and generate corresponding medical report descriptions. In addition, **E3D-GPT** introduced a large-scale 3D medical image dataset and a 3D medical image foundation model based on MAE He et al. (2022), which enhances the representation of visual information in the overall vision-language model.

While these methods have promising initial results, they still have significant limitations. They either fine-tune larger LLMs (E3D-GPT), use a simple linear projection of 3D radiology images (CT-AGRG), or fine-tune both the image encoder and text decoder (CT2Rep). This can lead to overfitting and a lack of proper alignment between visual and textual semantics.

### 2.2 VISION PROJECTOR IN LARGE VISION LANGUAGE MODELS (VLMS)

Vision projectors are modules designed to project the context of visual data in the same way as text, which helps align the image and text spaces. Early methods like LLaVA Liu et al. (2023) proposed

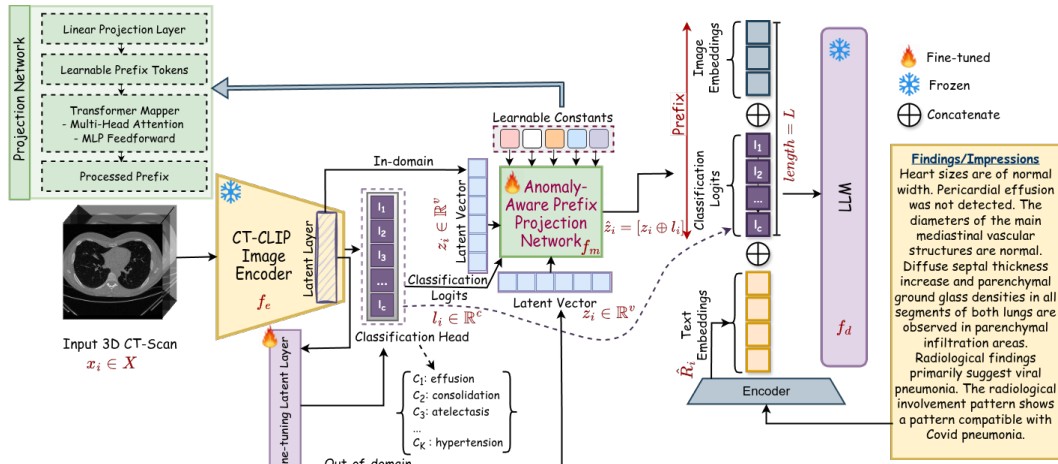

Figure 3: Overview of the proposed *RAD3D-Prefix* model. The model aligns the image encoder's output and the classification logits to the text embedding space via a lightweight projection network.

a simple feed-forward layer for this purpose, with promising initial results. Later, LLaMA 3.2 Meta (2024) introduced a cross-attention mechanism to tackle this alignment challenge.

In the context of medical imaging, the LLaVA-Med model Li et al. (2023a) uses a simple MLP projection layer, similar to the original LLaVA model Liu et al. (2023) . This approach works well for 2D images but isn't effective for 3D radiology images. For report generation, models like Med-2E3 Shi et al. (2024) and Red2RG Chen et al. (2024) use adapters to help projectors encode volumetric data, but they still have high computational costs. This is because they use two encoders to generate visual information, which significantly increases overhead. Like our proposed method, the R2GenGPT Wang et al. (2023) model uses a frozen LLM with a visual alignment layer. However, its alignment layer is a simple linear projection, which can lead to feature alignment issues.

To address these challenges, we propose **RAD3D-Prefix**, which uses a transformer-based, anomaly-aware prefix module. This module leverages prefix projection to effectively manage the differences between the image and text embedding spaces. This approach improves computational efficiency and provides better alignment, all while keeping the LLM frozen.

## 3 RAD3D-PREFIX

In this section, we describe our proposed **RAD3D-Prefix** model for CT report generation. Given a 3D CT-scan image $x_i \in \mathbb{R}^{C \times D \times H \times W}$ where C, D, H and W denote the channel, depth, height, and width, respectively, our framework aims to generate a patient-specific multi-sentence report impression/finding, $R$, in a clear and coherent manner. We leverage the strength of the vision foundation model and LLM. This approach allows for efficient report generation with only a few trainable parameters while still achieving enhanced outcomes and mitigating the above-mentioned limitations.

### 3.1 OVERALL FRAMEWORK

**RAD3D-Prefix** is presented in Fig. 3, comprising of a pretrained and frozen 3D image encoder $f_e$, a trainable transformer-based projection network $f_m$ and a frozen text decoder $f_d$. The encoder $f_e$ extracts the visual embedding that can be utilized by the decoder $f_d$ to generate a report $R_i = \{r_1, r_2, ..., r_N\} \in \mathbb{V}$ where $\mathbb{V}$ is the vocabulary and $N$ is the length of the report. During this process, the lightweight network $f_m$ tries to project the visual embeddings obtained from $f_e$ into the $f_d$'s token space using a prefix learning mechanism, thus ensuring alignment between the two modalities. For instance, given a dataset $D = (X, \mathcal{R})$, we extract the visual embeddings $\mathbf{z}_i$ for image $x_i \in X$ using $f_e$, which can be given as:

$$\mathbf{z}_i = f_e(x_i) \in \mathbb{R}^v, \tag{1}$$

To provide the model with crucial clinical context, our approach incorporates multi-anomaly classification logits, $\mathbf{l}_i \in \mathbb{R}^c$ (for $c$ anomaly labels). These logits are obtained from a separate, pre-trained classification head on the image encoder $f_e$. We directly fuse this high-level diagnostic information with the raw visual features by concatenating the image embeddings, $\mathbf{z}_i$, with the classification logits, $\mathbf{l}_i$. This concatenated vector, which combines both visual and clinical cues, is then used as input to our transformer-based projection network, $f_m$. The network transforms this combined input into a structured sequence of $L$-dimensional embedding vector, which act as a prefix for the LLM. Note that the **RAD3D-Prefix** is built on the *V-3* variant configuration, where the prefix integrates both image embeddings and classification logits, and the LLM remains frozen. The other two proposed variants, *V-1* and *V-2* are presented in the subsequent sections for comparative analysis.

## 3.2 VISUAL FEATURE EXTRACTION

Considering the complexity of 3D CT-scan images, we adopted a recently introduced CT-CLIP Hamamci et al. (2024a) image encoder that is pre-trained in a contrastive setting using a large number of 3D CT-focused image-text pairs. This encoder is based on the CT-ViT Hamamci et al. (2024c) architecture that processes the input $\mathbf{x} \in \mathbb{R}^{(240) \times 480 \times 480}$ as non-overlapping patches of shape $(10) \times 20 \times 20$, where 10 is the temporal patch size $t_p$ and 20 denotes spatial patch sizes $p_1$ and $p_2$. These patches are reshaped to $B \times T \times H \times W \times (C.t_p.p_1.p_2)$, where $B$ is the batch size and $T$ is the temporal patch count. A linear transformation is then applied to obtain frame embeddings of shape $B \times T \times \frac{H}{p_1} \times \frac{W}{p_2} \times v$, where $v$ is the final required latent representation dimension. The spatial transformer processes this reshaped tensor, maintaining the same size. Subsequently, it passes through a causal transformer, yielding a tensor of shape $\left(\frac{H}{p_1}.\frac{W}{p_2}\right) \times (B.T) \times v$. Such a combination of spatial and causal transformations ensures the retention of 3D information throughout the network. Finally, to obtain the latent representation, the embeddings obtained are processed through a linear layer to convert them into a vector of dimension $v$, where $v$ is set to 512. This latent vector is used as input for the projection network. For in-domain data samples, we freeze all parameters of $f_e$, whereas for out-of-domain data samples, we fine-tuned the last layer.

## 3.3 ANOMALY-AWARE PREFIX PROJECTION NETWORK

Our proposed model's main trainable component is a lightweight, transformer-based projection network, $f_m$. Its purpose is to align the image-derived embeddings with the LLM's token space. Unlike traditional projectors that only use visual features as a single token or sequence of tokens, our network is **anomaly-aware** that generates a semantically disentangled sequence of visual and clinical context and maps them into a sequence of LLM-style tokens. It leverages both a latent vector from the image encoder and a set of multi-label diagnostic classification logits to create a more comprehensive input. This design is motivated by the need to explicitly provide the model with high-level clinical priors, thus retaining critical diagnostic cues that might otherwise be lost.

To integrate the diagnostic information, we concatenate the multi-anomaly classification logits, $\mathbf{l}_i$, with the image features $\mathbf{z}_i$ obtained from $f_e$. This fusion creates a rich, combined representation, and can be defined as $\hat{\mathbf{z}}_i = [\mathbf{z}_i \oplus \mathbf{l}_i] \in \mathbb{R}^{v+c}$, where $\oplus$ denotes concatenation. Initially, these fused embeddings $\hat{\mathbf{z}}_i$ are projected into a structured sequence using a linear transformation. The obtained structured embeddings and a learnable constant serve as input for the projection network. The learnable constant extracts relevant information from the embeddings and adjusts the network to new data samples. The transformer layers within our projection network then operate in a self-attention setting to capture complex dependencies and optimize the representation. The output of this process is the final prefix, a fixed-length embedding sequence, called *anomaly-aware prefix*, which is then concatenated with the textual report embeddings $\hat{\mathbf{R}}_i = \{\hat{\mathbf{r}}_1, \hat{\mathbf{r}}_2, ..., \hat{\mathbf{r}}_N\}$, $\hat{\mathbf{r}}_j \in \mathbb{R}^h$, before being fed to the frozen LLM $f_d$. Here $h$ represents the LLM's hidden size. This design ensures that the model is conditioned not only on the raw visual content but also on clinically significant multi-abnormality markers, helping to generate more factual and diagnostically precise reports. The classification logits are kept soft and frozen during report generation to provide valuable clinical priors without introducing backpropagation errors into the classifier. This targeted, multi-modal input strategy ensures the LLM is well-conditioned to generate a factually accurate and diagnostically precise report.

**Training Objective:** The training objective is to optimize the lighweight, trainable projection network, $f_m$, while the image encoder $f_e$ and text decoder $f_d$ remain frozen. This objective is achieved by minimizing the negative log-likelihood of the target report sequence. The loss function is defined as:

$$\mathcal{L} = -\sum_{i=1}^{M} \sum_{j=1}^{N} log\, p_\theta(\hat{\mathbf{r}}_{i,j}|\hat{\mathbf{z}}_i, \hat{\mathbf{r}}_{i,1}, ..., \hat{\mathbf{r}}_{i,j-1}). \tag{2}$$

Here, $M$ is the number of reports in the dataset, and $N$ is the number of tokens in the $i$-th report. The term $p_\theta(\hat{\mathbf{r}}_{i,j}|.)$ represents the probability of predicting the $j$-th token of the $i$-th report, $r_{i,j}$, which is conditioned on the entire prefix, $\hat{\mathbf{z}}_i$ and all previously generated tokens. The parameters being optimized, denoted by $\theta$ belong exclusively to the projection network, $f_m$. The goal is to train this network to generate a prefix representation, $\hat{\mathbf{z}}_i$, that effectively conditions the frozen LLM ($f_d$) to produce the correct report tokens, $r_{i,j}$, in an autoregressive manner. This approach ensures that the LLM's vast, pre-trained knowledge is leveraged while the model learns to bridge the modality gap using only a minimal number of trainable parameters.

During the training phase, the projection network is optimized using the training loss $\mathcal{L}$ (Eq.2), constrained by the concatenated prefix and report embeddings ($\hat{\mathbf{z}}_i, \hat{\mathbf{r}}_{i,1}, ..., \hat{\mathbf{r}}_{i,j-1}$). In contrast, only the prefix projections ($\hat{\mathbf{z}}_i$) are used during evaluation, and the model generates the report tokens iteratively in an autoregressive manner, selecting each token based on the highest computed probabilities in the process. A detailed algorithm on projection network training is given in the Appendix (Algorithm 1). Also, further details on the classification module are discussed in the Appendix A.2.

### 3.4 REPORT DECODER

Report decoder $f_d$ is a pre-trained LLM, extensively trained on a large corpus of generic text data. We adopted the LLaMA-3.2-1B[2] model, originally designed to generate relevant text responses when given an input text prompt. This limits its ability to handle visual content, as they are encoded in a different format. Moreover, fine-tuning such large models could interfere with the model's generalizability and lead to suboptimal performance. Therefore, we used the decoder's pre-trained weights without fine-tuning to leverage its already captured rich and hierarchical text representations and avoid overfitting on small medical datasets. However, to align the medical semantic features and the textual data in the report, we used the projection network that reformulates visual cues and the semantics of the reports in a manner that LLaMA-3.2-1B can effectively interpret and process. The decoder $f_d$ iteratively generates the report tokens when given some cues in the form of prefix projections. This process can be defined as:

$$P(w_t \mid w_{<t}, \hat{\mathbf{z}}_{i,1:L_p}) = \text{softmax}(w_{out} \cdot f_d(w_{<t}, \hat{\mathbf{z}}_{i,1:L_p}), \tag{3}$$

where $L_p$ is the prefix length, $w_{<t}$ signifies the previously generated tokens and $\mathbf{w}_{out}$ is the output token projection matrix that maps LLaMA's final hidden state to vocabulary space.

## 4 EXPERIMENTS

### 4.1 CONFIGURATIONS

**Datasets:** We conducted experiments using two publicly available large datasets: CT-RATE Hamamci et al. (2024a) and INSPECT Huang et al. (2023) datasets. CT-RATE is used for in-domain evaluation as the CT-CLIP Hamamci et al. (2024a) encoder is originally pre-trained on this dataset, whereas INSPECT is used as an out-of-domain dataset. CT-RATE consists of 50,188 non-contrast chest CT volumes with 18 multi-abnormality labels. We used its official split, i.e., 47,149 and 3,039 scans for training and testing, respectively. The findings section is used for report generation.

INSPECT comprises CT-scans acquired from 19,402 patients, focused on pulmonary embolism. Therefore, we used it for out-of-domain evaluation. After removing redundant data, we used 17,730 and 3,506 scans for training and testing, respectively. Due to the unavailability of the official split

---

[2]https://huggingface.co/meta-llama/Llama-3.2-1B

| Model Name | Method | Avg. BLEU (1-4) | METEOR | Avg. ROUGE (1-2) | ROUGE-L | BERTScore-F1 |
|---|---|---|---|---|---|---|
| DistilGPT2 | Baseline[3] | 0.2660 | 0.4205 | 0.4416 | 0.4166 | 0.8832 |
| | V-1 | **0.2925** | **0.4271** | **0.4501** | **0.4281** | **0.8925** |
| GPT2 | Baseline[3] | 0.2476 | 0.4013 | 0.4380 | 0.4014 | 0.8810 |
| | V-1 | **0.3065** | **0.4366** | **0.4684** | **0.4315** | **0.8932** |
| GPT2-Medium | Baseline[3] | 0.2658 | 0.4321 | **0.4676** | **0.4314** | 0.8858 |
| | V-1 | **0.3037** | **0.4351** | 0.4672 | 0.4313 | **0.8931** |
| LLaMA-3.2-1B | Baseline[3] | 0.2637 | 0.4335 | **0.4677** | **0.4363** | 0.8864 |
| | V-1 | **0.3427** | **0.4467** | 0.4638 | 0.4062 | **0.8891** |

Table 1: Comparative analysis of baseline and variant V-1. Best values are in **bold**.

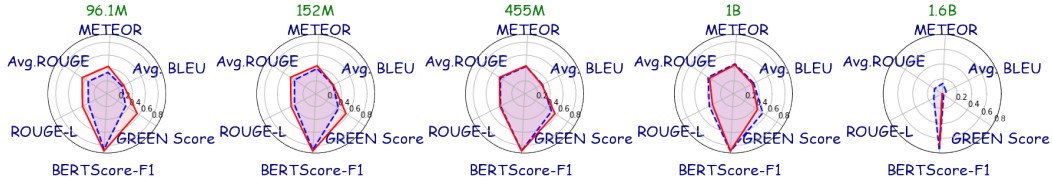

Figure 4: Radar graph plots demonstrating the impact of fine-tuning (solid lines) and freezing (dashed lines) parameters of variable-sized LLMs. From 96.1M to 1B graphs in sequence, the plots correspond to DistilGPT2, GPT2, GPT2-medium, LLaMA-3.2-1B, and BioGPT-Large, respectively.

details, a stratified sampling was performed based on 21 distinct anomalies identified as entities in the report impressions. Further details on dataset split preparation are given in Appendix A.1.

**Pre-processing and Training Details:** Inspired by CT-CLIP Hamamci et al. (2024a), each CT volume is resized for a uniform spacing, i.e., 0.75 mm in the x-axis and the y-axis, and 1.5 mm in the z-axis. All experiments are conducted using $480 \times 480 \times 240$ resolution with Hounsfield Units (H.U.) clipped to -1000 to 1000 range, followed by normalization. The experiments are performed on NVIDIA-A100 GPU using PyTorch framework, training all models for 10 epochs to ensure a fair comparison. The projection module is trained using an Adam optimizer with a learning rate of 2e-5.

| Method | LLM | No. of Parameters in LLM | Avg. BLEU (1-4) | METEOR | Avg. ROUGE (1-2) | ROUGE-L | BERTScore-F1 | GREEN Score (Clinical Efficacy) |
|---|---|---|---|---|---|---|---|---|
| Baseline[3] | BioGPT-Large | 1.6B | - | - | - | - | 0.6932 | - |
| R2GenGPT | BioGPT-Large | 1.6B | 0.0611 | 0.1385 | 0.1241 | 0.1284 | 0.7622 | 0.0254 |
| R2GenGPT | LLaMA-3.2-1B-Instruct | 1B | 0.2902 | 0.3762 | 0.39485 | 0.3468 | 0.8751 | 0.4120 |
| R2GenGPT | LLaMA-2-7b-chat-hf | 7B | 0.3523 | 0.4509 | **0.4640** | 0.4038 | **0.8886** | 0.5041 |
| **RAD3D-Prefix** | LLaMA-3.2-1B | 1B | **0.3637** | **0.4694** | 0.4256 | **0.4190** | 0.8883 | **0.5488** |

Table 2: Comparison with the state-of-the-art approach of vision and text embedding alignment and different sized LLM training. The analysis is divided into three categories: (a) LLM (both fine-tuned: baseline, row 1 and frozen: row 2) with specialized domain pre-training, (b) frozen LLM with the same parameter count (row 3) and (c) with higher parameter count using a conventional alignment approach (row 4). Values in **bold** denote the best outcome.

**Metrics:** We report results on four widely used NLG metrics, namely, BLEU (1-4) Papineni et al. (2002), METEOR Banerjee & Lavie (2005), ROUGE (1, 2, L) Lin (2004) and BERTScore-F1 Zhang et al. (2019). In addition, we adopt the GREEN score Ostmeier et al. (2024), which is specifically relevant to the medical domain. BLEU measures n-gram precision, with a penalty for short translations, whereas METEOR incorporates synonym matching and word order penalties. ROUGE focuses on n-gram recall, and the BertScore-F1 focuses on semantic similarity using a language model. The GREEN score focuses on the factual correctness of clinical information.

**Baseline:** We compared our method with various state-of-the-art techniques in terms of (a) vision-text alignment approach, (b) 3D medical image to report generation task, and (c) different sized LLMs adoption. We primarily compare our work with R2GenGPT Wang et al. (2023) due to its overall architectural similarities and emphasis on medical report generation. Although it was originally proposed for 2D images, we performed its training by feeding it our extracted visual latent embeddings. R2GenGPT uses a frozen LLM and, hence, to additionally compare with trainable

| Dataset | Method | BLEU-4 | METEOR | ROUGE-1 | BERTScore-F1 | GREEN |
|---------|--------|--------|--------|---------|--------------|-------|
| CT-RATE | E3D-GPT[†] | - | 0.4179 | 0.5260 | 0.8797 | - |
| | CT-AGRG[†] | 0.172 | 0.1960 | | 0.867 | - |
| | CT2Rep | **0.3212** | 0.4543 | **0.5899** | **0.8929** | 0.5247 |
| | RAD3D-Prefix | 0.2779 | **0.4694** | 0.5780 | 0.8894 | **0.5488** |
| INSPECT | CT2Rep | 5.74e-06 | 0.1207 | 0.1892 | 0.8629 | 0.2219 |
| | RAD3D-Prefix | **0.0344** | **0.2122** | **0.2268** | **0.8670** | **0.2400** |

Table 3: Comparison of state-of-the-art approach for 3D images to clinical report generation. [†]The results are from the original paper on the same test split. - denotes the unreported metrics.

| Dataset | Variant | Visual Prefix | Frozen LLM | Multi-label Classification Logits | Avg. BLEU (1-4) | METEOR | Avg. ROUGE (1-2) | ROUGE-L | BERTScore-F1 | GREEN (Clinical Efficacy) | No. of Trainable Parameters. |
|---------|---------|---------------|------------|-----------------------------------|-----------------|--------|------------------|---------|--------------|---------------------------|------------------------------|
| CT-RATE | V-1 | ✓ | ✗ | ✗ | 0.3315 | 0.4456 | 0.4402 | 0.3646 | 0.8826 | 0.4454 | 1.51B |
| | V-2 | ✓ | ✓ | ✗ | 0.3543 | 0.4604 | 0.4677 | 0.4190 | 0.8883 | 0.5428 | 279.09M |
| | V-3 | ✓ | ✓ | ✓ | **0.3637** | **0.4694** | **0.4716** | **0.4256** | **0.8894** | **0.5488** | 279.46M |
| INSPECT | V-1 | ✓ | ✗ | ✗ | **0.0984** | **0.2473** | 0.1586 | 0.1747 | 0.8505 | 0.1565 | 1.51B |
| | V-2 | ✓ | ✓ | ✗ | 0.0579 | 0.2028 | 0.1810 | 0.2189 | 0.8659 | 0.2355 | 279.09M |
| | V-3 | ✓ | ✓ | ✓ | 0.0657 | 0.2122 | **0.1889** | **0.2268** | **0.8670** | **0.2400** | 279.46M |

Table 4: Comparative analysis of the role of presence/absence of different features (Visual Prefix, Frozen LLM, and Multi-label Classification Logits) in the proposed RAD3D-Prefix model.

LLM, we created our baseline using basic clip-to-text decoder[3] architecture, using same visual embeddings but a conventional projection approach and fine-tunes the LLM during training.

We compared our proposed model with R2GenGPT using variable LLMs: domain-specialized LLM (BioGPT-Large Luo et al. (2022)), an LLM with the same number of parameters (LLaMA-3.2-1B-Instruct), and an LLM with higher parameters originally adopted in their study (LLaMA-2-7b-chat-hf). The other baseline is additionally used to validate our three proposed alignment module variants (*V-1*, *V2*, and *V-3*, shown in Fig. 2). Further, we performed a comparative analysis with recent work in the same domain. 3D CT report generation is understudied, with only a few methods available but without publicly available code for reproducibility (E3D-GPT, CT-ARG). Hence, we relied on reported values only. These works are closest in scope, along with CT2Rep.

## 4.2 RESULTS

This section compares the baseline approach and three proposed variants, namely, *V-1*, *V-2*, and *V-3*.

**Baseline vs. Variant *V-1*:** To assess the significance of including visual embedding as a prefix (Variant *V-1*), we compared its report generation outcomes with that obtained using baseline (without prefix). As shown in Table 1, the majority of metrics favour the variant *V-1* across four different LLMs with varying parameter count. The overall results indicate an increase of 9.96% to 29.95% in Avg. BLEU, 0.69% to 8.80% in METEOR, and a maximum of 6.94% and 7.50% improvement in Avg. ROUGE (1-2) and ROUGE-L, respectively. Similarly, an increase of about 1.39% is observed in BERTScore-F1. The reported improvements highlight the superior performance of variant *V-1* over the baseline, underscoring the importance of the visual prefix in enhancing report generation.

**Variant *V-1* vs. Variant *V-2*:** This comparison discusses the key concept of this work, i.e., the impact of fine-tuning and freezing parameters of smaller ($< 1B$) and larger ($\approx 1B+$) LLMs. Fig. 4 clearly illustrates that models with parameter sizes ranging from 96.1M to 455M exhibit a performance decline across various metrics, including the GREEN Score, when adopted as frozen. Note that both fine-tuned (variant *V-1*) and frozen (variant *V-2*) models with 96.1M to 1B parameters utilize the same projection module, involving visual embeddings. In contrast, the last model, BioGPT-Large, is trained using the standard baseline approach for fine-tuning, whereas the R2GenGPT model deploys it as a frozen module. The model exhibited severe overfitting when fine-tuned.

We can conclude two findings from this comparison: (a) With the increase in parameter number, the performance gap between the frozen and fine-tuned models starts to decrease. Further increasing the parameters (beyond 1B) improved the frozen model's performance compared to its fine-tuned counterpart. (b) Our proposed projection network outperforms the conventional mapping criteria followed in the existing work, as demonstrated by the degraded performance of the baseline models

---

[3]https://github.com/fkodom/clip-text-decoder

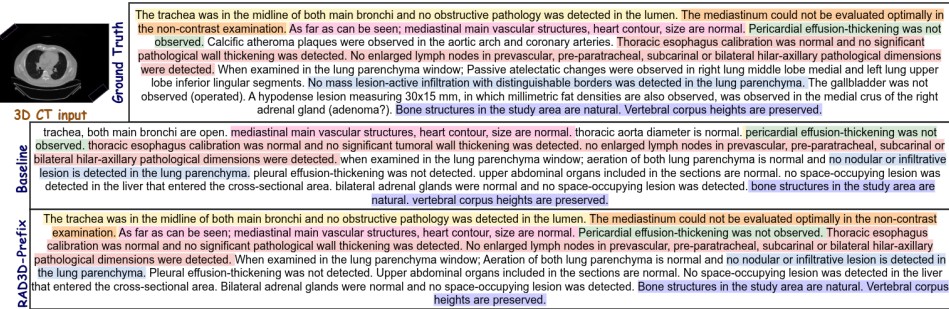

Figure 5: Qualitative example of the baseline and RAD3D-Prefix. Matching sentence pairs are highlighted in the same color.

in the last graph plot. Although the overall trend between the frozen and fine-tuned models remains consistent, a clear distinction in performance is evident. We further compare the *V-3* variant in the ablation study section below. Also, qualitative results are shown in Fig. 5. More qualitative samples and classification outcome details of *V-3* are given in the Appendix (Fig. 8).

## 4.3 COMPARISON WITH STATE-OF-THE-ART METHODS

In this section, we compare our approach with state-of-the-art methods for medical report generation. We focus on models with similar (i) architectural frameworks (R2GenGPT) and (ii) methodological objectives (3D image to report generation) to ensure a meaningful comparison. Table 2 presents a comparative analysis of our approach against the first category models with (a) specialized domain LLM, (b) same parameter count LLM, and (c) higher parameter count LLM. We used BioGPT-Large Luo et al. (2022) as a specialized domain LLM in our baseline setting, i.e., a conventional mapping network with fine-tuning. The model experiences significant overfitting, and hence, the results are not appropriate and are not reported in the table. We further used the same model with R2GenGPT, i.e., as a frozen module and demonstrated better performance than its baseline version.

In addition to the specialized LLMs, we trained R2GenGPT with two more LLMs, LLaMA-3.2-1B-Instruct and LLaMA-2-7b-chat-hf. We selected LLaMA-3.2-1B-Instruct to conduct a fair comparison with the same version and size of LLaMA. An instruction variant of LLaMA was selected to serve a similar purpose to the original LLM (LLaMA-2-7b-chat-hf) used by R2GenGPT. The results indicate that our model clearly outperforms the same-sized LLM, i.e., LLaMA-3.2-1B-Instruct, achieving a GREEN score of 0.5488. Furthermore, even when a larger LLM is deployed in a similar setting, our model outperformed it on several metrics and performed comparably on others. For a fair comparison, we also replaced our transformer-based projection network with a linear layer (similar to R2GenGPT) for which details are discussed in Appendix A.5.

We further evaluated our approach against existing 3D medical image-to-report generation methods. Table 3 presents results from several recent techniques applied to the same dataset. Our method, **RAD3D-Prefix**, outperforms most existing techniques on the CT-RATE dataset, with the exception of CT2Rep on certain NLG metrics. However, our model's superiority is evident in its clinical relevance, as demonstrated by a higher GREEN score compared to CT2Rep. Furthermore, **RAD3D-Prefix** significantly outperformed CT2Rep on the out-of-domain INSPECT dataset, underscoring its generalizability and robustness in real-world scenarios.

## 4.4 ABLATION STUDY AND OUT-OF-DOMAIN PERFORMANCE

We performed an ablation study to assess the significance of different concepts introduced in *RAD3D-Prefix*, especially the anomaly-aware projection module, as shown in Table 4. Additionally, INSPECT, an out-of-domain dataset is utilized to assess the impact of various features when applied to a dataset from a different distribution. It is evident that incorporating each feature leads to improvements across most metrics. The GREEN score, which is considered the most relevant metric from a clinical perspective, demonstrates a 23.22% improvement in the variant *V-3* setting compared to variant *V-1*. Similarly, using the INSPECT dataset, a 53.36% improvement is observed.

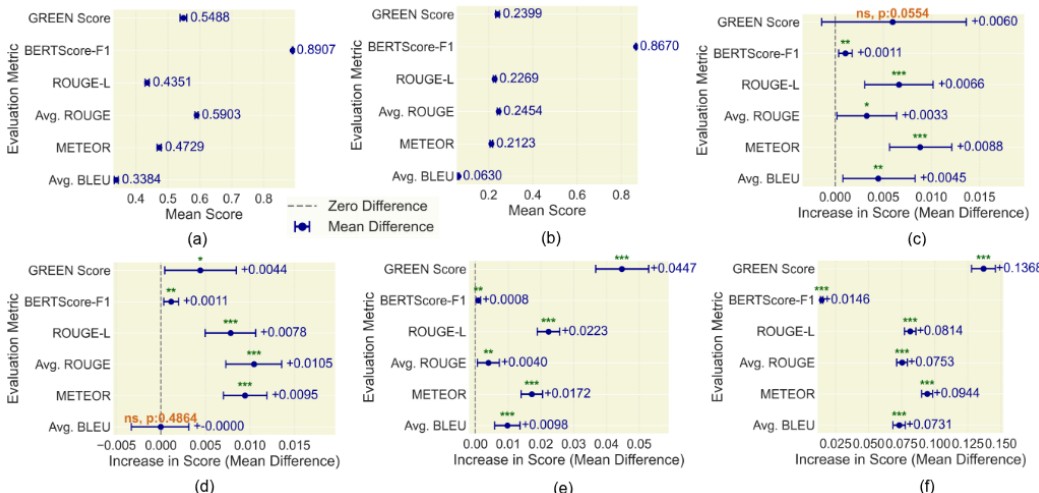

Figure 6: The forest plots show the performance means/mean differences and 95% CIs for RAD3D-Prefix on (a) CT-RATE and (b) INSPECT, and its comparisons with (c–f) Variant V2 on both datasets, R2GenGPT w/ LLaMA-2-7b-chat-hf, and RGenGPT w/ LLaMA-3.2-1B-Instruct. Asterisks indicate significance thresholds: $^*p < 0.05$, $p < 0.01$, $^*p < 0.001$; "ns" denotes $p \geq 0.05$.

### 4.5 STATISTICAL SIGNIFICANCE EVALUATION

To quantify the statistical stability of each metric and to evaluate the significance of performance improvements relative to baseline methods, we performed bootstrapping with 5,000 iterations. For each resample, metric values and their differences from other models were recomputed, providing a mean and mean difference estimate and their 95% confidence interval (CI). Fig. 6 (a) and Fig. 6 (b) demonstrate our model's consistent and stable performance, as indicated by the narrow bootstrapped 95% CI across all metrics using the CT-RATE and INSPECT datasets, respectively. In addition, we compared *V-2* and *V-3* variants across the two datasets (Fig. 6 (c) and (d)) to assess the significance of introducing classification logits into our proposed projection network. We observed that five out of six metrics illustrate statistically significant improvements ($p < 0.05$), signifying robust and generalized efficacy of the **RAD3D-Prefix** that hold across independent datasets and evaluation dimensions, thereby validating the significance of the anomaly-aware prefix. In Fig. 6 (c), although the GREEN score improvement does not meet the significance threshold but the marginal p-value ($p = 0.0554$) suggests that the improvement is borderline meaningful and may warrant further investigation. Similarly, in Fig. 6 (d), the mean difference in Avg. BLEU is minimally negative (-0.000031), suggesting that *V-2* achieved a marginally higher score. However, since the $p$-value is high ($p = 0.4864$), this marginal difference is not statistically significant. Further, we compared our model against the baseline, R2GenGPT with a larger (Fig. 6 (e)) and similar-sized LLM (Fig. 6 (f)). The consistent performance gain with all metrics meeting the significance criterion ($p < 0.05$) highlights the robustness of our method. Consequently, it is evident that the proposed anomaly-aware projection network is statistically robust.

## 5 CONCLUSION

In this paper, we present RAD3D-Prefix model, with a *lightweight anomaly-aware prefix projection* module for the generation of clinical reports from 3D CT scans. The lightweight projection module effectively learns an alignment between the image features (latent as well as classification logits) and the text features. Despite having fewer trainable parameters than the SoTA, the proposed model outperforms the existing model on four different evaluation metrics by 20.8% to 33.2%. Additionally, we compare different variants of the proposed work to highlight scenarios where one variant can be preferred compared to another variant.

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

# A  APPENDIX

## A.1  DATASET DETAILS

### A.1.1  CT-RATE

The CT-RATE dataset consists of 25,692 non-contrast chest CT volumes, which are expanded to 50,188 using various reconstruction methods. These scans come from 21,304 unique patients and are accompanied by corresponding radiology text reports, 18 multi-abnormality labels, and metadata. During training, samples from 20,000 patients (47,149 scans) are used, while the remaining samples from 1,304 patients (3,039 scans) are reserved for testing. The associated radiology reports are segmented into four sections: clinical information, technique, findings, and impression. However, only the findings section is used to train the report generation model. For the classification task, the publicly available 18 multi-abnormality labels are used.

### A.1.2  INSPECT

The INSPECT dataset has 23,248 scans focusing mainly on pulmonary embolism. After removing some redundant data, we used 17,730 and 3,506 scans for training and testing, respectively. As the dataset is not accompanied by additional abnormality labels and an official split, ReXKG Zhang et al. (2024) is used to extract entities representing abnormalities. These extracted entities are further utilized for two purposes: (a) stratified train and test dataset split, and (b) multi-anomaly classification to obtain classification logits. After applying ReXKG, the obtained entities are sorted in descending order based on their frequency of occurrence. Entities occurring more than 1,000 times across both training and testing sets are selected for multi-anomaly classification. Also, anomalies with the same Concept Unique Identifier (CUI) are combined, with their frequencies summed accordingly. For example, *emboli, embolus, embolism* have a common CUI *C1704212*, therefore, their occurrence frequencies are aggregated under one entity *"Embolism"*. A list of these anomalies with their occurrence count is shown in Table 5. This distribution is used to create a training and test split of the INSPECT dataset via stratified sampling based on the occurrence of anomalies in the reports.

| Anomaly | Training Set | Test Set |
|---|---|---|
| Embolism | 3691 | 739 |
| Opacities | 3788 | 758 |
| Nodules | 4296 | 859 |
| Disease | 2592 | 518 |
| Atelectasis | 2916 | 583 |
| Infection | 2744 | 549 |
| Thickening | 2526 | 505 |
| Effusions | 4608 | 922 |
| Edema | 1775 | 355 |
| Aspiration | 1543 | 308 |
| Consolidation | 1828 | 366 |
| Mass | 1403 | 281 |
| Pneumonia | 1367 | 273 |
| Metastatic | 1283 | 256 |
| Lymphadenopathies | 1557 | 311 |
| Hypertension | 1048 | 209 |
| Metastases | 1137 | 227 |
| Lesions | 1648 | 330 |
| Malignancy | 1070 | 214 |
| Node | 850 | 169 |
| Others | 3302 | 661 |

Table 5: Distribution of anomaly counts in the INSPECT dataset for both the training and test sets.

---

**Algorithm 1** Anomaly-Aware Projection Network Training

---

**Input:** Image embeddings $\mathbf{z}_i$ obtained using $f_e$, text embeddings $\hat{\mathbf{R}}_i$ and classification logits $\mathbf{l}_i$
**Ensure:** Frozen LLaMA-3.2-1B with $f_m$ projection network for report generation
1: **for** each batch $(\mathbf{z}_i, \hat{\mathbf{R}}_i, \mathbf{l}_i) \in D$ **do**
2:     Concatenate $\mathbf{z}_i$ with $\mathbf{l}_i$: $\hat{\mathbf{z}}_i \leftarrow \text{concat}(\mathbf{z}_i, \mathbf{l}_i)$, where $\hat{\mathbf{z}}_i \in \mathbb{R}^{v+c}$
3:     Construct prefix mask: $\mathbf{p}_{mask} \leftarrow \mathbf{1}^{B \times L_p}$, where $B$ is the batch size and $L_p$ is the prefix length (10 in our case)
4:     Concatenate prefix mask with $\hat{\mathbf{R}}_i$ attention mask (obtained using LLaMA-3.2-1B tokenizer): mask $\leftarrow \text{concat}(\mathbf{p}_{mask}, \hat{\mathbf{R}}_i.\text{attention\_mask}, \dim = 1)$
5:     Pass $\hat{\mathbf{z}}_i$ through the linear projection layer in $f_m$: $\mathbf{E}_{proj} \leftarrow \mathbf{W}\hat{\mathbf{z}}_i + \mathbf{b}$, where $\mathbf{E}_{proj} \in \mathbb{R}^{L_p \times h}$, $\mathbf{W} \in \mathbb{R}^{(v+c) \times (L_p.h)}$, $\mathbf{b}$ is the bias term and $h$ is the LLM's hidden size.
6:     Reshape($\mathbf{E}_{proj}, B, L_p, h$), $\mathbf{E}_{proj_{reshape}} \in \mathbb{R}^{B \times L_p \times h}$
7:     Define the learnable prefix constant: $\mathbf{p}_c$, $\mathbf{p}_c \in \mathbb{R}^{B \times L_p \times h}$
8:     Concatenate $\mathbf{E}_{proj_{reshape}}$ with $\mathbf{p}_c$: $\mathbf{S} \leftarrow \text{concat}(\mathbf{E}_{proj_{reshape}}, \mathbf{p}_c, \dim=1)$
9:     Pass $\mathbf{S}$ through $K$ transformer layers (here $K = 8$). For each transformer layer, process through multi-head attention and an MLP feedforward network.
10:     Compute loss $\mathcal{L}$ (Eq. 2)
11:     Backpropagate and update only $f_m$ parameters
12: **end for**
13: **return** fine-tuned model for report generation

---

## A.2 CLASSIFICATION MODULE AND RESULTS

To obtain the multi-abnormality classification logits, we fine-tuned the last layer of the CT-CLIP encoder and added a classification layer. (Fig. 3). Note that the same CT-CLIP encoder used for CT scan feature extraction is employed for this purpose; however, as an independent module, separate from the report generation pipeline. For the CT-RATE dataset, we used its original weights from Hamamci et al. (2024a) since the CT-CLIP encoder is pre-trained with CT-RATE and for the IN-SPECT dataset, its last layer is fine-tuned. The training process involves binary cross-entropy loss with class-wise weights to address class imbalance. A multi-label classification is performed featur-

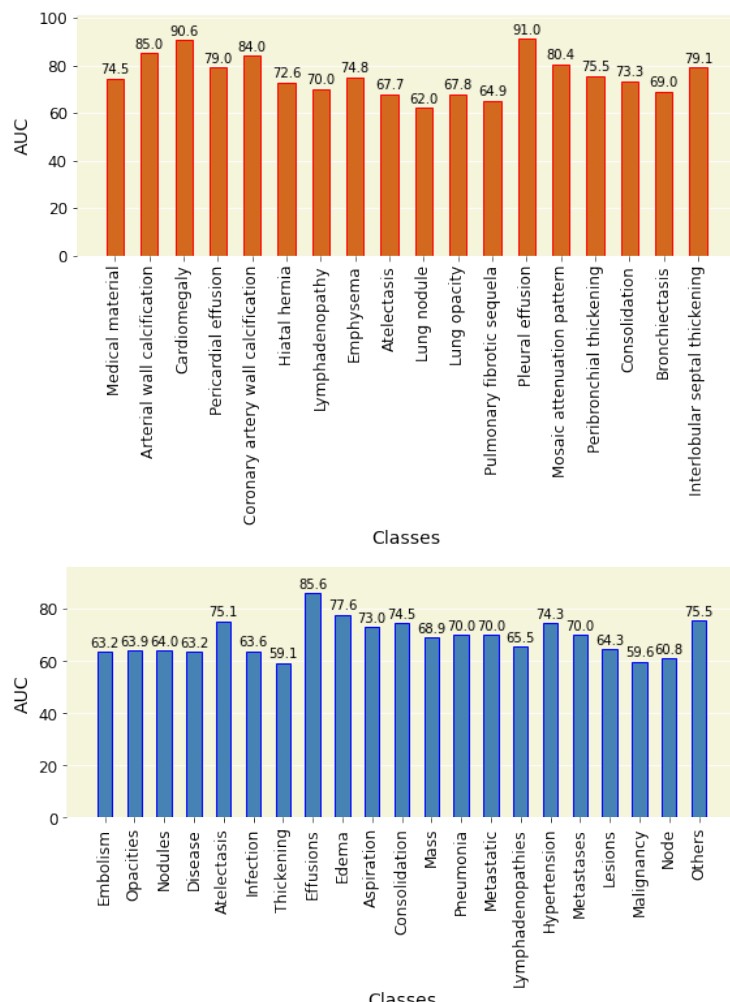

Figure 7: Classification results on 18 and 21 multi-abnormality labels of the CT-RATE (top) and INSPECT (bottom) dataset, respectively.

ing the 18 and 21 abnormalities of the CT-RATE and INSPECT datasets, respectively. Therefore, the output dimension equals the number of findings (CT-RATE: 18, INSPECT:21). We concatenate these 1D logits with image embeddings, and this adds anomaly-specific priors to guide generation. The classification results are shown in Fig. 7. These results demonstrate strong anomaly detection, capturing clinical patterns that enhance report generation, particularly for critical conditions (e.g., effusion: 91.0%, cardiomegaly: 90.6%).

| Method | LLM | No. of Parameters in LLM | F1-RadGraph | RaTEScore |
|---|---|---|---|---|
| R2GenGPT | BioGPT-Large | 1.6B | 0.0208 | 0.3338 |
| R2GenGPT | LLaMA-3.2-1B-Instruct | 1B | 0.2310 | 0.6370 |
| R2GenGPT | LLaMA-2-7b-chat-hf | 7B | 0.2783 | 0.6746 |
| **RAD3D-Prefix** | LLaMA-3.2-1B | 1B | **0.2884** | **0.6830** |

Table 6: Comparison with the state-of-the-art approach of vision and text embedding alignment and different sized LLM training based on two more clinically-oriented metrics.

A.3 QUALITATIVE RESULTS

More qualitative samples are given in Fig. 5. It can be seen that the baseline model is overfitted on the dataset, generating the same result regardless of the input. In contrast, our model performed better with input-specific output covering most of the anomalies. Some failure cases of our model involved content where abnormalities were accompanied by specific measurements. Nevertheless, our model correctly identified critical conditions, including effusions and hiatal hernia.

Fig. 9 illustrates reports predicted by the three variants, *V-1*, *V-2*, and *V-3*. It can be observed that RAD3D-Prefix, based on *V-3* variant, roduces the report most aligned with the radiologist-annotated ground truth. It is the only variant that correctly identifies "COVID-19 pneumonia", "no obstructive pathology", "ground-glass opacity with correct location". Following *V-3*, *V-2* performed better than V-1, capturing "ground-glass opacity" but with incorrect location, while *V-1* fails to detect it entirely.

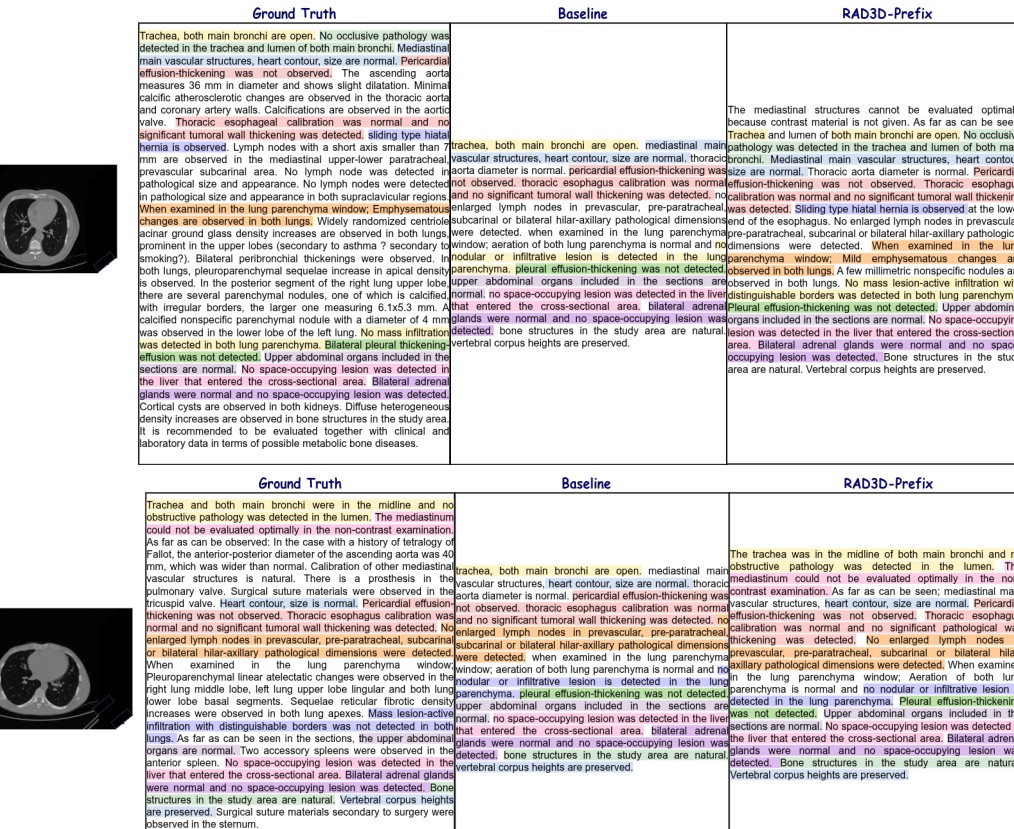

Figure 8: More qualitative examples of the baseline and our proposed method. Matching sentence pairs are highlighted in the same color.

A.4 GREEN SCORE DEFINITION AND COMPUTATION DETAILS

GREEN Score Ostmeier et al. (2024) is a metric for radiology report generation that uses regular expressions to parse errors in generated reports and to identify matched findings. The score can be calculated as:

$$\text{GREEN Score} = \frac{\#\ \text{matched findings}}{\#\ \text{matched findings} + \sum_{i \in \text{sig. errors}} \#\ \text{errors}_i}, \tag{4}$$

where a "matched finding" is a clinical observation present in both the generated and reference reports. " errors" correspond to findings whose omission or hallucination would plausibly impact

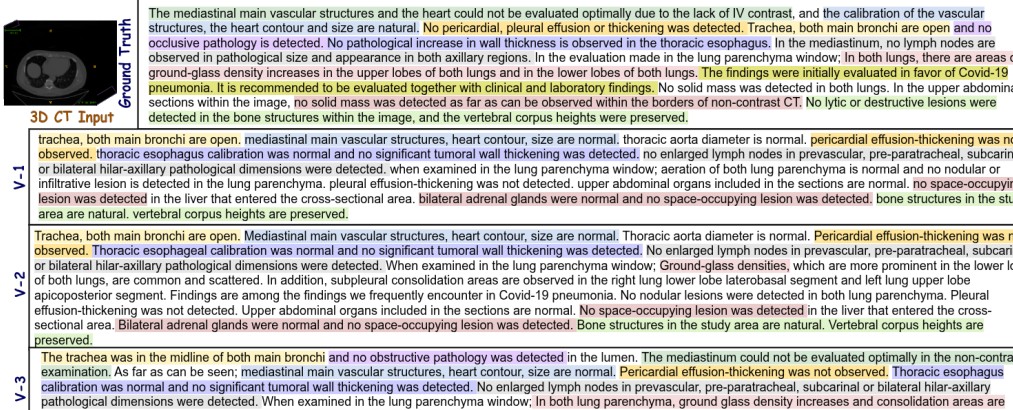

Figure 9: Qualitative sample comparing outcomes of the three variants, *V-1*, *V-2*, and *V-3*.

| Method | LLM (1B Frozen Para.) | Projection (Trainable Para.) | Avg. BLEU | METEOR | ROUGE-L | BERTScore-F1 | GREEN (Clinical Efficacy) | F1-RadGraph | RaTEScore |
|---|---|---|---|---|---|---|---|---|---|
| **R2GenGPT** | LLaMA-3.2-1B -Instruct | Linear Layer (1.05M) | 0.2902 | 0.3762 | 0.3468 | **0.8751** | 0.4120 | 0.2310 | 0.6370 |
| **Proposed** | LLaMA-3.2-1B | Linear Layer (1.05M × 10) | **0.3165** | **0.4161** | **0.3824** | 0.8714 | **0.4826** | **0.2559** | **0.6435** |

Table 7: Using a linear layer in the projection network instead of the transformer-based network and comparing the setting with R2GenGPT.

clinical decision-making. If no matched findings are present, the GREEN score is defined to be zero. Sample GREEN Summaries for *V-2* and *V-3* variants are shown in Fig. 10. These two variants are compared specifically because *V-3* incorporates classification logits while *V-2* does not, enabling a direct comparison of the impact of logits on clinical correctness. Across 3039 test cases, *V-3* produces 253 additional matched findings (1.43% improvement) compared to *V-2*, corresponding to approximately one additional finding every 12 cases. Given the sparsity and clinical importance of radiology findings, this improvement is clinically meaningful.

### A.5 PROJECTION NETWORK PARAMETER COUNT COMPARISON

To support the effectiveness of our proposed framework, we present results replacing our transformer mapping with a linear layer (see Table 7). This variant still uses our proposed prefix design, resulting in increased parameter count from 1.05M (linear) to 1.05M×10 (linear+prefix), still significantly less than 279.5M (original transformer+prefix). Replacing our transformer mapper (279.5M) with linear layer cuts parameters 26× with only a 0.047 BLEU ↓ (still outperforming R2GenGPT). This shows gains stem from the prefix mechanism, not just increased model size. 10× extra parameters above account for the prefix length (*our core proposal*).

### A.6 ANALYSIS OF PREFIX LENGTH AND LAYERS VS. PERFORMANCE AND TRAINABLE PARAMETERS

We performed an additional ablation study with different prefix lengths (2,5,7, 10, and 14) and the number of layers in the transformer architecture of the projection network (2, 4, 8, and 16) to analyze tradeoffs between performance and computational overhead. The results are shown in Fig. 11. It can be observed that increasing prefix length significantly improves performance up to a threshold, after which gains saturate. The most significant improvement occurs between prefix lengths 2 and 5 with gains above 60% across five metrics. Beyond prefix length 5, the improvements are marginal despite increased parameter count. Further, Fig. 11 (b) shows that while increasing the number of layers dramatically increases trainable parameters, there is no measurable performance gain across

**Reference:** Mediastinal vascular structures and cardiac examination were not evaluated optimally because of the lack of IV contrast. As far as can be seen; Calibration of vascular structures, heart contour and size are natural. Pericardial, pleural effusion was not detected. Trachea, both main bronchi are open and no occlusive pathology is detected. No pathological increase in wall thickness was observed in the thoracic esophagus. In the mediastinum, no lymph nodes were detected in pathological size and appearance in both axillary regions. When examined in the lung parenchyma window; Plaque-like linear calcification in the pleura is observed in the apical segment of the right lung upper lobe, adjacent to the mediastinum, and there are sequelae parenchymal changes in the adjacent lung parenchyma. There was no finding in favor of pneumothorax in both lungs. No active infiltration or mass lesion was observed in both lungs. Ventilation of both lungs is natural. In the upper abdominal sections within the image, no pathology was detected as far as it can be observed within the borders of non-contrast CT. No lytic or destructive lesions were observed in the bone structures within the image. Vertebral corpus heights are preserved.

**V-2 Prediction:** The mediastinal main vascular structures and the heart could not be evaluated optimally due to the lack of IV contrast, and the calibration of the vascular structures, the heart contour and size are natural. No pericardial, pleural effusion or thickness increase was observed. Trachea, both main bronchi are open and no occlusive pathology is detected. No pathological increase in wall thickness was observed in the thoracic esophagus. No lymph node was detected in the mediastinum in pathological size and appearance. When examined in the lung parenchyma window; No active infiltration or mass lesion was detected in both lung parenchyma. Pleuroparenchymal sequelae density increases were observed in the apex of both lungs. In the upper abdominal sections within the image, no solid mass was detected as far as can be observed within the borders of non-contrast CT. No lytic or destructive lesions were observed in the bone structures in the study area.

**V-3 Prediction:** No lymph node was observed in the supraclavicular fossa, axilla and mediastinum in pathological size and appearance. Heart dimensions and compartments appear natural. Pericardial effusion was not detected. Calibrations of mediastinal major vascular structures are normal. No lymph node was observed in the mediastinum in pathological size and appearance. Trachea, both main bronchi are open. No occlusive pathology was detected in the trachea and both main bronchi. No pathological wall thickness increase was observed in the esophagus within the sections. No enlarged lymph nodes in prevascular, pre-paratracheal, subcarinal or bilateral hilar-axillary pathological dimensions were detected. When examined in the lung parenchyma window; Aeration of both lung parenchyma is normal and no nodular or infiltrative lesion is detected in the lung parenchyma. Pleural effusion-thickening was not detected. Upper abdominal organs included in the sections are normal. No space-occupying lesion was detected in the liver that entered the cross-sectional area. Bilateral adrenal glands were normal and no space-occupying lesion was detected. Bone structures in the study area are natural. Vertebral corpus heights are preserved.

**GREEN SUMMARY (V-2)**

The candidate report is largely consistent with the reference report, with most findings matching. However, there are a few discrepancies. The candidate report mentions "Pleuroparenchymal sequelae density increases" which is not mentioned in the reference report. Also, the candidate report does not mention the "Plaque-like linear calcification in the pleura" and "vertebral corpus heights are preserved" which is present in the reference report.

[Clinically Significant Errors]:
(a) False report of a finding in the candidate: 1. Pleuroparenchymal sequelae density increases.
(b) Missing a finding present in the reference: 2. Plaque-like linear calcification in the pleura; Vertebral corpus heights are preserved.
(c) Misidentification of a finding's anatomic location/position: 0.
(d) Misassessment of the severity of a finding: 0.
(e) Mentioning a comparison that isn't in the reference: 0.
(f) Omitting a comparison detailing a change from a prior study: 0.

[Clinically Insignificant Errors]:
(a) False report of a finding in the candidate: 0.
(b) Missing a finding present in the reference: 0.
(c) Misidentification of a finding's anatomic location/position: 0.
(d) Misassessment of the severity of a finding: 0.
(e) Mentioning a comparison that isn't in the reference: 0.
(f) Omitting a comparison detailing a change from a prior study: 0.

[Matched Findings]:
8. Mediastinal vascular structures and cardiac examination were not evaluated optimally due to the lack of IV contrast; Calibration of vascular structures, heart contour, and size are natural; Trachea, both main bronchi are open; No occlusive pathology is detected in the thoracic esophagus; No lymph node is detected in the mediastinum in pathological size and appearance; No active infiltration or mass lesion is detected in both lung parenchyma; No pathology is detected in the upper abdominal sections within the image limits of the non-contrast CT; No destructive lesions are detected in the bone structures within the image limits of the non-contrast CT.
**green_score: 0.7272727272727273**
(a) False report of a finding in the candidate: 1
(b) Missing a finding present in the reference: 2
(c) Misidentification of a finding's anatomic location/position: 0
(d) Misassessment of the severity of a finding: 0
(e) Mentioning a comparison that isn't in the reference: 0
(f) Omitting a comparison detailing a change from a prior study: 0
**Matched Findings: 8**

**GREEN SUMMARY (V-3)**

The candidate report is largely consistent with the reference report, but there are a few discrepancies. The candidate report does not mention the plaque-like linear calcification in the pleura of the right lung upper lobe, which is a clinically significant error. It also does not mention the sequelae parenchymal changes in the adjacent lung parenchyma, which is another clinically significant error. The candidate report also mentions the normal condition of the bilateral adrenal glands, which is not mentioned in the reference report, but this is not a clinically significant error as it does not affect the patient's diagnosis or treatment.

[Clinically Significant Errors]:
(b) Missing a finding present in the reference: 2. Plaque-like linear calcification in the pleura of the right lung upper lobe; Sequelae parenchymal changes in the adjacent lung parenchyma.

[Clinically Insignificant Errors]:
(e) Mentioning a comparison that isn't in the reference: 1. Normal condition of the bilateral adrenal glands.

[Matched Findings]:
12. No lymph node was observed in the supraclavicular fossa, axilla and mediastinum; Heart dimensions and compartments appear natural; Pericardial effusion was not detected; No pathological increase in wall thickness was observed in the thoracic esophagus; No lymph node was observed in the mediastinum in pathological size and appearance; Trachea, both main bronchi are open; No occlusive pathology was detected in the trachea and both main bronchi; No pathological increase in wall thickness was observed in the thoracic esophagus; No lymph node was observed in the mediastinum in pathological size and appearance; No nodular or infiltrative lesion was detected in the lung parenchyma; No pathological increase in wall thickness was observed in the thoracic esophagus; No lymph node was observed in the mediastinum in pathological size and appearance.
**green_score: 0.8571428571428571**
(a) False report of a finding in the candidate: 0
(b) Missing a finding present in the reference: 2
(c) Misidentification of a finding's anatomic location/position: 0
(d) Misassessment of the severity of a finding: 0
(e) Mentioning a comparison that isn't in the reference: 0
(f) Omitting a comparison detailing a change from a prior study: 0
**Matched Findings: 12**

Figure 10: Samples of GREEN Summary for V-2 and V-3 variants.

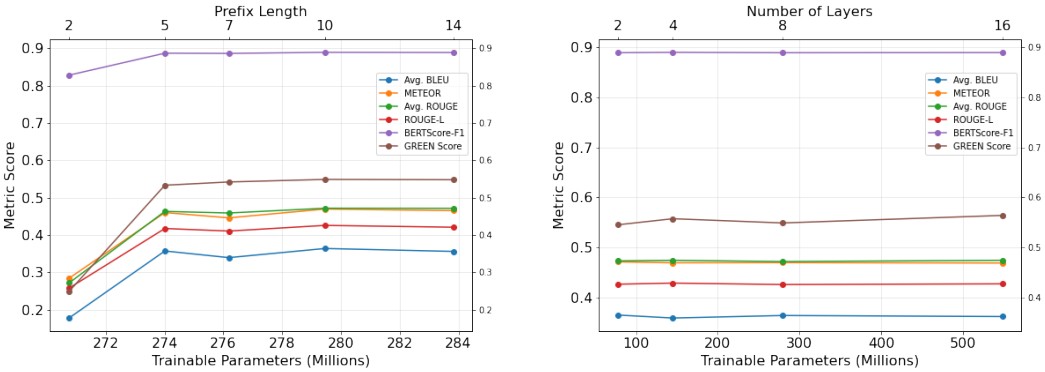

Figure 11: Performance analysis with respect to increasing trainable parameters, influenced by modifications to (a) Prefix Length, (b) Number of Layers in the transformer-based projection network.

NLG metrics. Although a slight improvement is observed in the GREEN metric, the gain is not proportional to the increase in parameters.

## A.7 ANALYSIS PERFORMANCE WITH DIFFERENT LLMS WITHIN RAD3D-PREFIX

We compared our model with a larger LLM, DeepSeek-R1-Distill-LLaMA-8B and the results are reported in Table 8. It can be observed that across different LLM configurations, RAD3D-Prefix consistently outperforms the R2GenGPT baselines. Using LLaMA-3.2-1B, RAD3D-Prefix achieves the

highest Avg. BLEU, METEOR, and the best GREEN score (0.5488), indicating improved clinical relevance. When paired with the larger DeepSeek-R1-Distill-LLaMA-8B, RAD3D-Prefix further improves ROUGE metrics and BERTScore-F1, showing that the framework effectively scales with stronger LLMs. Although R2GenGPT performs better with a 7B LLM compared to its 1B version, RAD3D-Prefix with only a 1B LLM already surpasses or matches the 7B baseline across most metrics, highlighting the efficiency and scalability of the proposed approach.

| Method | LLM | No. of Parameters in LLM | Avg. BLEU (1-4) | METEOR | Avg. ROUGE (1-2) | ROUGE-L | BERTScore-F1 | GREEN Score (Clinical Efficacy) |
|---|---|---|---|---|---|---|---|---|
| R2GenGPT | LLaMA-3.2-1B-Instruct | 1B | 0.2902 | 0.3762 | 0.39485 | 0.3468 | 0.8751 | 0.4120 |
| R2GenGPT | LLaMA-2-7b-chat-hf | 7B | 0.3523 | 0.4509 | 0.4640 | 0.4038 | 0.8886 | 0.5041 |
| **RAD3D-Prefix** | LLaMA-3.2-1B | 1B | **0.3637** | **0.4694** | 0.4256 | 0.4190 | 0.8883 | **0.5488** |
| **RAD3D-Prefix** | DeepSeek-R1-Distill-LLaMA-8B | 8B | 0.3405 | 0.4459 | **0.4704** | **0.4339** | **0.8909** | 0.5443 |

Table 8: Analyzing performance of the RAD3D-Prefix framework using different LLM configuration

## A.8 UMAP Visualization of Pathology Patterns

To inspect the learned representation in the *V-2* and *V-3* variants, we projected the embeddings onto the three components and visualized them from multiple viewpoints (Fig.12). In all plots, the data occupy a thin, curved manifold rather than filling the 3D space, indicating a strong intrinsic low dimensionality of chest CT appearances. The manifold is single and connected, consistent with a continuous spectrum of disease rather than discrete clusters. Radiologic labels are heavily intermixed, as expected in a multi-label setting, but show local enrichment along different parts of the manifold (e.g., bronchiectasis and interlobular septal thickening toward one extremity, pleural effusion and consolidation along central segments).

Across *V-2* and *V-3* variants, the global topology is stable, however, *V-3* yields a more compact S-shaped ribbon with slightly reduced thickness, reflecting a more structured organization of the latent space while preserving the overall disease continuum.

V-2: In Fig. 12a, the blue/cyan ribbon is looser, less structured, and appears more scattered, suggesting noisy embedding. It has more scattered noise points floating around the structure.

V-3: In Fig. 12b, the blue/cyan ribbon is tighter and more clearly defined. This indicates the V3 model's prefix is more confident and consistent in encoding the features of common pathologies. It has a denser core with fewer isolated "outlier" points far from the main body.

The above analysis demonstrates that the anomaly-aware mechanism successfully creates unique vectors for structurally important diseases.

## A.9 Disclosure

LLM has been used to polish writing and make minor edits.

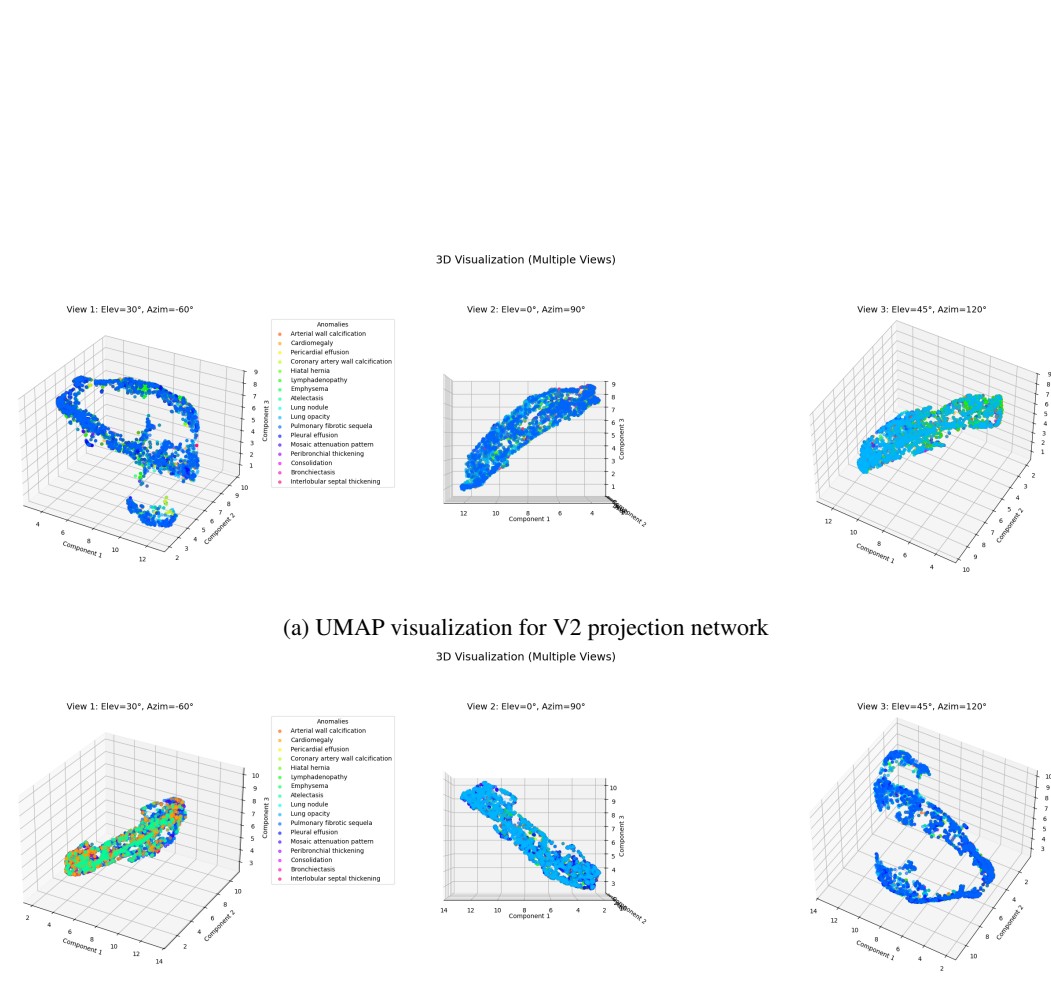

Figure 12: UMAP visualizations and qualitative examples across different projection networks.

