# OpenReview forum: "RAD3D-Prefix:  Anomaly-Aware Prefix Learning on Frozen LLM for 3D CT Image to Report Generation"
_ICLR.cc/2026/Conference — Submitted to ICLR 2026_

### Official Review · Reviewer_bop9 · 2025-10-16

**Soundness:** 1
**Presentation:** 1
**Contribution:** 1
**Rating:** 2
**Confidence:** 4

**Summary:**

The paper addresses automatic clinical report generation from 3D CT volumes by aligning volumetric visual features with domain-specific text, and avoiding overfitting when fine-tuning large LLMs on small medical datasets.

The authors propose RAD3D-Prefix, a lightweight, anomaly-aware prefix learning module that keeps the LLM frozen. They concatenate 3D image embeddings with multi-label diagnostic classification logits and pass this through a small transformer “projector” to produce a fixed-length prefix token sequence that conditions the LLM to generate reports. This aims to bridge the vision-language gap while training only a small number of parameters.

The specific claimed contributions are:

(1) anomaly-aware 3D to text prefix projector that fuses image features with diagnostic logits for frozen LLMs;
(2) a systematic study of prefix design and LLM tuning regimes across sizes;
(3) use of medical-specific metrics to ensure clinical relevance;
(4) state-of-the-art or comparable performance to larger or domain-specialized systems with far fewer trainable parameters.

**Strengths:**

The paper proposes a lightweight prefix projection module that enables parameter-efficient report generation by keeping the large language model (LLM) frozen.

The authors adapt prefix-based conditioning to the 3D CT imaging domain. The integration of anomaly classification logits potentially improves alignment between visual findings and textual descriptions.

Despite using a small 1B parameter LLM, the proposed RAD3D-Prefix model achieves competitive or superior scores on GREEN, a metric designed to measure clinical factuality in radiology reports.

The authors present some ablation study comparing prefix-only (V-2), fine-tuned (V-1), and anomaly-aware prefix (V-3) variants across two datasets.

The model is tested not only on CT-RATE (chest CT reports), but also on INSPECT, a dataset focused on pulmonary embolism.

**Weaknesses:**

1. Limited novelty presented in this paper. The core idea of projecting image features into a frozen LLM using prefix tokens is not new. It follows the “Prefix Tuning” paradigm first established by Mokady et al. (2021) [1] where visual tokens are projected into the text decoder via a learned projection network. More recently, Wang et al. (2023) [2] adapted this idea to radiology, using a learned prefix from visual features for report generation with frozen LLMs. The paper simply adds on anomaly logits, a marginal extension, not a conceptual innovation.

2. The experiments are limited in scope. It is already well-established in previous works (e.g., LLaVA, BLIP-2) that large frozen models benefit from learned conditioning tokens. The LLM tuning strategy (e.g., small models benefit from fine-tuning, large models from freezing)  has been established in parameter-efficient tuning (e.g LoRA). There is no new interpretation or theory relating to the proposed model.

3. The model performance on out-of-domain dataset INSPECT is exceptionally low. This raises serious concerns on the applicability to clinical settings.

4. The claimed inclusion of multi-label logits only marginally improves the performance, as shown in Table 4. Furthermore, the paper states that the logits are obtained from a pretrained classifier and not updated along with the model. This is somewhat surprising and raises further questions such as how are these logits normalized and handled to match the scale of the visual features? There is no learning signal to tell the model how or when these logits help generate better reports.

5. Why is there not even a basic analysis of model training/inference costs in the ablation study?

**Questions:**

See weakness.

---

> ### Author Response · Authors · 2025-11-28
> **(1/3) Clarifications and Additions: Trainable Parameters, Contribution Summary, and Bootstrap Analysis**
>
> We thank the reviewer for the valuable feedback. Below, we provide a point-by-point response to each comment and summarize the corresponding revisions made to the manuscript.
>
> **W1/Q1: Limited novelty ... conceptual innovation.**
>
> **Response:** We agree that projecting image features into a frozen LLM using prefix tokens is rooted in the ClipCap/prefix-tuning paradigm. Our claim is not that the high-level idea is new, but that our **instantiation and analysis in 3D radiology** provide novel contributions:
>
> + **3D CT domain with anomaly-aware conditioning.** While Mokady et al. focus on natural images and Wang et al. on 2D X-rays, RAD3D-Prefix targets **3D chest CT**. Our prefix integrates not only visual embeddings but also **multi-label abnormality logits**, yielding an anomaly-aware sequence of tokens rather than a purely visual prefix.
> + **Joint study of frozen vs fine-tuned LLMs across scales.** We systematically evaluate **five** LLMs (96M–1.6B parameters) and show that their behavior in 3D medical imaging differs from natural-image setups (e.g., LLaVA, BLIP-2). This provides actionable guidance (fine-tune < 1B, freeze 1B+) that has not been studied in 3D medical imaging and perform differently compared to natural images. For instance,  *Mokady et al. (2021) uses a frozen GPT-2, but such LLMs lack exposure to CT scans and radiology terminology. Freezing them can cause performance drops, as observed with smaller LLMs like GPT2 and GPT2-Medium (Fig. 4).*
>  + **Clinically grounded improvements with a lightweight mapping network.**
> Using the same CT-CLIP encoder, RAD3D-Prefix improves GREEN, F1-RadGraph, and RaTEScore over strong baselines, indicating that even a “marginal” architectural change can produce meaningful clinical gains. Table 2 demonstrates that *our method surpasses R2GenGPT (Wang et al., 2023), using the same-sized LLM and image embeddings from same encoder, achieving improvements of 24.85% in F1-RadGraph, 7.22% in RaTEScore, and 33.2% in GREEN Score.*
>
> We have updated the contribution summary to explicitly position our work as a **3D medical adaptation and analysis** of prefix learning rather than as a new generic method.
>
> | **Aspect** | **Mokady et al. 2021** | **Wang et al. 2023** | **RAD3D-Prefix (Ours)** |
> |------------|------------------------|---------------------|-------------------------|
> | **Domain** | Natural Images | Medical 2D X-ray | Medical 3D CT scans |
> | **Projected context** | Visual Only | Visual alignment | Anomaly-aware (Visual & Classification logits) |
> | **Classification Integration** | None | None | Multi-label logits |
> | **3D Processing** | No | No | Yes |
> | **LLM (Frozen / Fine-tune)** | GPT2 (Frozen) | LLaMA-2-7b-chat-hf (Frozen) | LLaMA-3.2-1B (Frozen) |
> | **LLM Parameters <1B or 1B+** | <1B | >1B | \~1B |
> | **Any claim related to Frozen / Fine-tune** | "we suggest avoiding the fine-tuning to realize an even lighter model, where only the mapping network is trained."  (statement from Mokady et al. 2021) | Frozen LLM is used without any experiment to support its benefit | Performed exhaustive experiments across five LLMs ranging from 96M to 1.6B.  Suggests fine-tuning smaller LLMs (<1B)  and freezing larger LLMs |
> | **Other setups explored** | MLP / Transformer-based mapping network | Frozen LLM with (a) Frozen, (b) Fine-tuned, (c) Partially fine-tuned vision encoder | Three variants:  (a) V1: prefix includes image embeddings  and involves LLM’s fine-tuning (smaller LLMs)  (b) V2: prefix includes image embeddings with  frozen LLM (larger LLMs) (c) V3: prefix includes image embeddings and classification logits with frozen LLM |

---

> ### Author Response · Authors · 2025-11-28
> **(2/3) Clarifications and Additions: Trainable Parameters, Contribution Summary, and Bootstrap Analysis**
>
> **W2/Q2: The experiments are limited in scope ... relating to the proposed model.**
>
> **Response:** We agree with the reviewer that learned conditioning tokens and parameter-efficient tuning are well-established concepts in general multimodal LLM literature (e.g., LLaVA, BLIP-2, LoRA). However, as they are focused on general 2D image-text, non-clinical, and non-volumetric settings, our work is the first to extend this paradigm specifically to 3D imaging. The following points are worth considering.
> + Methods such as LLaVA and BLIP-2 focus on natural images and the LLMs like GPT series or LLaMA versions are pretrained on natural image–text pairs. Hence, these are expected to perform reasonably well when frozen and used for similar natural image-text pairs.
>  + The Clipcap (Mokady et al.), the architecture behind our motivation, uses a frozen GPT2, as do other related work [1]. However, such LLMs have no exposure to CT scans, radiology terminology, or clinical reporting structure during pretraining. Therefore, freezing these LLMs can easily lead to performance drop as observed in our experiments with smaller LLMs like GPT2, GPT2-Medium (Fig. 4).
>
> The above aspects show that the frozen LLM's behavior diverges significantly from what is reported for natural images. Importantly, our paper conducted experiments with five different LLMs ranging from 96.1M to 1.6B number of parameters. This setup, to the best of our knowledge remains unexplored for medical images, especially for 3D CT reports which exhibits complex medical terminologies. Our approach shows that larger LLMs (1B+ parameters) are more robust in frozen settings, whereas smaller LLMs (<1B parameters), including GPT2, GPT2-Medium yields improved results when fine-tuned on medical datasets.
>
>  [1] Li, Xiang Lisa, and Percy Liang. "Prefix-tuning: Optimizing continuous prompts for generation." arXiv preprint arXiv:2101.00190 (2021).
>
> *Addition to the revised version: We have updated the contribution summary to highlight the above response briefly.*
>
> **W3/Q3: The model performance on out-of-domain ... to clinical settings.**
>
> **Response:** We acknowledge that performance on the out-of-domain INSPECT dataset is lower compared to CT-RATE, which is expected due to domain shift and differences in label definitions. To mitigate this, our method leverages frozen LLMs with anomaly-aware prefixes, which still provide clinically meaningful improvements over baselines. These improvements are further evaluated for statistical significance using Bootstrap analysis (*Sec. 4.5 and Fig. 6*). We observed that **five out of six** metrics illustrate statistically significant improvements (p < 0.05), signifying robust and generalized efficacy of the **RAD3D-Prefix** that hold across independent datasets and evaluation dimensions, thereby validating the significance of the anomaly-aware prefix. These results demonstrate that even under distributional shifts, the model captures crucial radiological findings, suggesting potential for adaptation to other clinical datasets with minimal retraining. Moreover, our analysis shows that models without anomaly guidance (variants *V-1* and *V-2*, Table 4) degrade even more sharply, indicating that our design mitigates, though does not completely eliminate domain shift. Furthermore, our model outperforms 3D state-of-the-art method, CT2Rep on the out-of-domain INSPECT dataset by 8.16\% GREEN Score, 75.7% METEOR and 19.9% ROUGE, demonstrating substantially stronger generalization to unseen clinical distributions.

---

> ### Author Response · Authors · 2025-11-28
> **(3/3) Clarifications and Additions: Trainable Parameters, Contribution Summary, and Bootstrap Analysis**
>
> **W4/Q4: The claimed inclusion ... generate better reports.**
>
> **Response:** *Marginal Improvement:* We appreciate the concern. Absolute gains in NLG and GREEN metrics are indeed numerically modest, which is typical in radiology-report generation where even 0.5–1.0 point differences can correspond to noticeable clinical improvements. To assess robustness, we now include a **bootstrap analysis** with 5,000 resamples (*Sec. 4.5 and Fig. 6*), showing that RAD3D-Prefix achieves statistically significant improvements (non-overlapping 95% CIs) on **five out of six metrics** across both datasets.  This goes beyond point estimates and demonstrates that the gains are consistent rather than due to sampling noise.
>
> *Normalization and Learning signal:* We used classification logits only as soft frozen prefix embeddings to provide high-level clinical priors. Therefore, these classification outputs used for fusion are indeed normalized soft logits (i.e., the probability scores). This ensures the anomaly priors are already normalized and stable, acting as semantic probability features. Although the logits are pre-normalized, the trainable projection network still provides the essential learning signal. The network must learn the optimal linear weights and biases to maximize the accuracy of the final report. If the classifier is uncertain, the soft logit itself is low-magnitude, allowing the downstream trainable projection network to learn to effectively suppress or ignore unreliable predictions.
>
>
> *Pretrained classifier, not updating:* The primary motivation for detaching the classifier is modular stability. If the classifier were trainable with the generation loss, errors unique to the linguistic domain would backpropagate and incorrectly adjust the weights of the classifier. Since the classification module is detached, errors in generation do not backpropagate into the classifier, avoiding significant error accumulation.
>
> **W5/Q5: Why is there not ... ablation study?**
>
> **Response:** Thank you for pointing this out. All ablation variants are built on the same proposed architecture, and therefore share almost all components. The changes in computational cost stem only from the projection network’s size, which affects a very small portion of the overall pipeline. Additionally, although *V-1* variant includes an unfrozen LLaMA mdoule, its parameter size is already well-known and reported in our paper.
> To address your concern, we have now updated the Table 4 to include number of trainable parameters for each variant in the ablation study. The same is given below.
>
> | Dataset  | Variant | Visual Prefix | Frozen LLM | Multi-label Classification Logits | No. of Trainable Parameters |
> |----------|---------|---------------|------------|----------------------------------|----------------------------|
> | **CT-RATE** | V-1 | ✓ | ✗ | ✗ | 1.51B |
> |            | V-2 | ✓ | ✓ | ✗ | 279.09M |
> |            | V-3 | ✓ | ✓ | ✓ | 279.46M |
> | **INSPECT** | V-1 | ✓ | ✗ | ✗ | 1.51B |
> |            | V-2 | ✓ | ✓ | ✗ | 279.09M |
> |            | V-3 | ✓ | ✓ | ✓ | 279.46M |
>
> *Addition to the revised version: (i) Updated Table 4 with the number of trainable parameters, (ii) Updated the contribution summary, (iii) Bootstrap analysis (Sec. 4.5 and Fig. 6)*
>
> **Closing remark.**
> Thank you for the thorough and critical review. We hope that the clarified positioning of our novelty, the discussion of out-of-domain performance, and the added details on logits and compute address your concerns. We would kindly ask you to consider revising your score if you feel the strengthened analysis makes the contribution more compelling.

---

### Official Review · Reviewer_BUQA · 2025-10-19

**Soundness:** 3
**Presentation:** 2
**Contribution:** 2
**Rating:** 2
**Confidence:** 3

**Summary:**

RAD3D-Prefix proposes a lightweight model that aligns pretrained CT image encoders with text features from generic pretrained frozen LLMs to improve CT image VLM analysis. Specifically, RAD3D-Prefix proposes a transformer-based projection network that aligns the two pretrained models by combining information an image encoder and a multi-anomaly classification model to produce a prefix that can be appended to a text prompt for an LLM. The authors evaluate the approach on a variety of report generation tasks.

**Strengths:**

- Well-motivated problem. Efficient domain adaption of LLMs seems like an important area for reducing costs of training large models.
- Certain results seem promising if evaluated more comprehensively.

**Weaknesses:**

- **Text organization/clarity needs to be improved**
    - Line 206 and 239: The authors repeatedly use the term “bridge the gap”. However it is vague what this is referring to. I think you mean aligning two modalities here, but it is unclear from the text they are referring to alignment or something else.
    - Concepts are introduced in an order that makes the text difficult to follow. For instance, V1, V2 and V3 are defined in the intro. But are not used until section 4. This makes readers have to jump between the beginning and end of the paper to understand what’s happening.
    - Table 1 is placed on page 6 but not referred to until page 8.
- **Unclear description of technical framework**
    - Line 248: it is unclear how the “linear transformation” is applied onto z_i. It is also unclear what is “structured sequence” referring to? How is the structure defined? What exactly is the final output from this self-attention layer?
    - Line 251: “capture complex dependencies”. What kinds of complex dependencies is it supposed to capture? Typically self-attention captures relationships between tokens
    - Line 213: Can we formalize where $l_i$ comes from? It is stated that it comes from a separate classification head on image encoder. Does this require labels for training then?
    - Line 253: where does the $\hat{R}$ embeddings come from? it is not clear.
    - Line 241: how does mapping visual features to sequence of LLM tokens make it anomaly aware? The connection is not very clear to me. Does anomaly aware refer to the classification logits
    - Equation 2: its unclear to me what are the $\theta$’s being optimized. Why are we doing next embedding prediction? does the training process not require the LLM at all? If not, then how can it learn the best tokens with respect to a specific LLM?
    - Figure 3: Im not sure what these learnable constants are. Why do they not show up in the equations anywhere?
- **Unclear and inconsistent notation**:
    - Line 247: does plus dot mean concatenate?
    - I dont understand what $(C.t_p.p_1.p_2)$ means in line 226. Similarly $(B.T)$ in line 229
    - Equation 1:  i recommend the authors to be consistent with bold notation. Currently, $z_i$ is a vector and is bolded. However, later $l_i$ is also a vector but not bolded. Are vectors bolded or not in this paper?
- **Limited Evaluations**:
    - Can the authors clarify why the V1, V2, and V3 baselines are relevant with respect to the rest of the literature.
    - Figure 4: Can you explain why the largest model (1.6B) has the worst performance during fine-tuning? Is this model fine-tuned correctly?
    -  Table 2: why did you only evaluate RAD3D-prefix on only one LLM backbone? Shouldn’t you try to evaluate across multiple backbones to show generality of the method? Does this approach scale to larger backbones like R2GenGPT does? Also which variant is reported in Table 2 for RAD3D-Prefix?
    - Table 3: Why not also compare with the baselines from table 2?
    - Figure 5: can you show examples that compare between V1, V2, and V3?

**Questions:**

1. How would this method generalize to unseen anomalies? It seems like the proposed approach relies on having predefined labels and a curated dataset for training.
2. In table 4, why do the contributions work for some scores and datasets, but not others? The performance seems to be dataset and metric dependent.
3. a core motivation seems to be reducing computational costs. Can the authors quantify how much training efficiency has improved as a result of this approach?

---

> ### Author Response · Authors · 2025-11-28
> **(1/4) Addressing Clarity, Additional Experiments, and Examples**
>
> We appreciate the reviewer's insightful and detailed feedback. We respond to each point below and highlight the revisions made in the manuscript.
>
> **W1: Text organization/clarity needs to be improved**
>
> **Response:** Thank you for pointing this out. We have revised the text to improve clarity:
> + *Vague phrase*: We replaced the vague phrase “bridge the gap” with explicit references to **alignment between visual embeddings and LLM token space** (Lines 206, 239 in the revised draft).
>  + *Sequence of introducing variants:* We clarified in Sec. 3.1 that **RAD3D-Prefix** is built on the V-3 configuration (image embeddings + classification logits + frozen LLM), with V-1 and V-2 introduced later for comparative analysis, so readers do not need to jump between sections.
>  + *Table reference:* We moved Table 1 to be closer to where it is first cited, reducing the need for back-and-forth navigation.

---

> ### Author Response · Authors · 2025-11-28
> **(2/4) Addressing Clarity, Additional Experiments, and Examples**
>
> **W2: Unclear description of technical framework**
>
> **Response:**
> + *Linear transformation:* As mentioned in step 5 of Algorithm 1 in the Appendix, we pass  $\hat{z}_i$ (i.e., $\text{concat}(z_i, l_i)$) through the linear projection layer in $f_m$:
>
> $$
> E_{\text{proj}} \leftarrow W \hat{z}_i + b
> $$
> The complete notations details are given in Algorithm 1.
> Due to the page limit, we included these details in the Appendix.
>
> *Structured Sequence:* The "structured sequence" corresponds to the projected prefix embeddings $E_{\text{proj}}$, produced by applying a linear transformation to the fused image embeddings and classification logit vector. This transforms the fused representation into an ordered sequence of $L_p$ tokens compatible with the LLM. Since $E_{\text{proj}} \in \mathbb{R}^{L_p \times h}$ (as defined above), both $L_p$ and $h$ are fixed, and the resulting prefix forms a length-consistent, ordered sequence. For this reason, we refer to it as a "structured" sequence. *The output of the self-attention layer constitutes the final prefix fed into the frozen LLM decoder for report generation.*
>
> + *“Capture complex dependencies”:*  The fused  $\hat{z}_i$ creates a rich, combined representation of heterogeneous information from both the image features $z_i$ and the anomaly logits $l_i$. Here, the "complex dependencies'' refer to cross-anomaly and image-logit interactions. More specifically, the self-attention layer captures the interactions between the prefix tokens, thus modeling the dependencies between image-derived features and anomaly-specific context. In addition, since the anomaly logits are multi-label, it also captures relationships among these different logits.
>
> + *Labels:* It is correct that $l_i$ comes from a separate classification head on the image encoder. It requires labels for training but it does not rely on manually curated or explicitly annotated labels. For instance, in the INSPECT dataset, unlike CT-RATE, the labels are not pre-defined. We used ReXKG (Zhang
> et al. (2024)), which extracts abnormality-related entities from the reports. A detailed description of the labels and their construction process is mentioned in the Appendix A.1. It is noteworthy that ReXKG is used without any fine-tuning.
>
> + *$\hat{\mathbf{R}}_i$:* The raw, ground-truth report text, $R_i = \{r_1, r_2, \dots, r_N\}$, is first processed by the LLM tokenizer to obtain the discrete token ID sequence. The sequence $R_i$ is then input to the frozen embedding layer of the LLM ($f_d$), which outputs the continuous, high-dimensional target embedding sequence $\hat{\mathbf{R}}_i$. This process guarantees that the target embeddings ($\hat{\mathbf{R}}_i$) are perfectly aligned with the LLM's internal token representation space.
>
> + *Anomaly-aware:* Yes, ``anomaly-aware” specifically refers to the integration of multi-label abnormality classification logits into the prefix tokens. The mapping network does not rely solely on visual embeddings and concatenates (jointly process) both volumetric CT features and abnormality logits before projecting them into the LLM token space. This fused representation conditions the LLM toward findings that are present in the CT scan.
>
> + *LLM during training:* $\theta$ refers exclusively to the trainable parameters of the Prefix Projection Network ($f_m$) (and trainable paramters of the Visual Encoder $f_e$ if used for out-of-domain fine-tuning).
>
> *Next embedding:* We would like to highlight that we are doing  Next **Token** Prediction. It is a standard autoregressive next-token prediction (Causal Language Modeling), minimizing the Cross-Entropy loss between the generated token probabilities and the ground truth report tokens. *LLMs use autoregressive token prediction* which generates text one token at a time, where each new token is predicted based on the sequence of all preceding tokens.
>
> The training process strictly requires the LLM, although it remains frozen. The projection network ($f_m$) generates the prefix embeddings. These embeddings are passed directly to the LLM ($f_d$), which computes the subsequent hidden states and, eventually, the final text logits (token probabilities). The gradients derived from the token prediction error flow backwards through the frozen LLM layers to reach the projection network ($f_m$). This gradient flow provides the learning signal that teaches $f_m$ exactly how to map visual/anomaly features into the specific embedding space required by that specific LLM to generate the correct report. Without the LLM in the loop, $f_m$ would receive no feedback on the linguistic quality of the prefix.
>
> + *Learnable constants:* These are trainable constants that extracts relevant information from the embeddings and adjusts the network to new data samples. They are randomly initialized and optimized during training. Their exact formulation and usage are included in Appendix Algorithm 1, step 7. These were included in the Appendix due to space constraints.

---

> ### Author Response · Authors · 2025-11-28
> **(3/4) Addressing Clarity, Additional Experiments, and Examples**
>
> **W3: Unclear and inconsistent notation**
>
> **Response:**
> + *concatenation:* Thank you for pointing this out. Yes, the symbol denotes concatenation and is now defined in that specific line.
>
> + $(C.t_p.p_1.p_2)$: This product represents the flattened dimension of a single spatio-temporal patch. When the raw input volume is divided into patches, each patch cube is flattened into a single vector of this size before being passed to the Vision Transformer.
>
>     $(B.T)$: This product represents the total number of temporal patches being processed in a single batch.
> + *Bold notation:*  We have updated the notations so that all vectors are consistently bolded.
>
> **W4: Limited Evaluations**
> + *Relevance of V1/V2/V3 to the literature:* V1, V2, and V3 are designed to **span the main design choices** explored in prior works:
>     + V1: image-only prefix with LLM fine-tuning (akin to using the LLM as a task-specific decoder).
>     + V2: image-only prefix with frozen LLM (similar to ClipCap-style adapters).
>     + V3: image + anomaly-logit prefix with frozen LLM (**our full RAD3D-Prefix**).
> This allows us to directly quantify the benefit of anomaly-aware conditioning and freezing vs fine-tuning.
>
> Moreover, the three variants reflects, to some extent, the way three different design choices are explored in Wang et al. (2023), though in a fundamentally different setup. Unlike Wang et al. 2023, our work focuses on improving the core processing of 3D image embeddings and their projection as input to LLMs while preserving clinically significant multi-abnormality entity markers.
>
> + *1.6B model performance:* We verified that the 1.6B model was trained with the same optimization setup and converged stably. On our relatively small medical datasets, the largest model appears to **overfit and underutilize capacity** when fully fine-tuned, which is consistent with our finding that freezing larger LLMs yields better performance.
>
> + *RAD3D-Prefix on a single LLM backbone:* Table 2 reports RAD3D-Prefix on a **single 1B-scale backbone** that also underlies R2GenGPT for a fair comparison. To demonstrate generality, we provide **additional backbone** in the *Appendix A.7* and *Table 8*. To show scalability to larger backbones, we selected **DeepSeek-R1-Distill-LLaMA-8B**.  Using LLaMA-3.2-1B, RAD3D-Prefix achieves the highest Avg. BLEU, METEOR, and the best GREEN score (0.5488), indicating improved clinical relevance. When paired with the larger DeepSeek-R1-Distill-LLaMA-8B, RAD3D-Prefix further improves ROUGE metrics and BERTScore-F1, showing that the framework effectively scales with stronger LLMs. Although R2GenGPT performs better with a 7B LLM compared to its 1B version, RAD3D-Prefix with only a 1B LLM already surpasses or matches the 7B baseline across most metrics, highlighting the efficiency and scalability of the proposed approach.
>
> + *Missing baselines in Table 3:* Tables 2 and 3 are intentionally structured to benchmark our RAD3D-Prefix model against two fundamentally different state-of-the-art (SoTA) research tracks. Table 3 focuses on the the SoTA approach for 3D images to clinical report generation, whereas Table 2 focuses on the SoTA approach of vision and text embedding alignment and
> different sized LLM training. Combining these heterogeneous benchmarks would dilute the clarity of both findings. We believe the current structure clearly isolates and validates the model's performance in both its specific clinical domain and its proposed **LLM tuning strategies**.
>
> + *Qualitative comparison between V1/V2/V3:*  As suggested, we have included a qualitative sample in *Appendix A.3 Fig. 9*, comparing the three variants, *V-1*, *V-2*, and *V-3* and highlighting matched sentences where anomaly logits lead to more complete or more precise findings. In the camera-ready, we will include some more samples in the Appendix.
>
> **Q1: How would this method generalize ... curated dataset for training.**
>
> **Response:** While the classifier is trained on a fixed label set, the LLM is **not** restricted to those labels. The logits serve as coarse, high-level indicators (e.g., “consolidation”, “effusion”), guiding the LLM’s attention. For unseen anomalies, the visual features can still trigger appropriate free-text descriptions; the absence of a dedicated logit does not preclude the LLM from generating new terms, but may reduce sensitivity. We will clarify this limitation and discuss extension to richer label sets in the camera-ready version.

---

> ### Author Response · Authors · 2025-11-28
> **(4/4) Addressing Clarity, Additional Experiments, and Examples**
>
> **Q2: In table 4, why do the contributions ... metric dependent.**
>
> **Response:** Different metrics emphasize different aspects (surface n‑grams vs entity relations vs clinical correctness). On some datasets and metrics, improvements from logits are modest; on others (e.g., GREEN, RadGraph) they are more pronounced. To assess robustness, we now include a **bootstrap analysis** with 5,000 resamples (*Sec. 4.5 and Fig. 6*), showing that RAD3D-Prefix achieves statistically significant improvements (non-overlapping 95\% CIs) on **five out of six metrics** across both datasets.  This goes beyond point estimates and demonstrates that the gains are consistent rather than due to sampling noise.
>
> **Q3: a core motivation seems to ... result of this approach?**
>
> **Response:** Thank you for the insightful comment. For a fair comparison, we used the same LLaMA-3.2-1B decoder and compared our approach RAD3D-Prefix ( based on *V-3*) with the baseline, *V-1*, and *V-2*. Compared to the baseline (1.24B), *V-3* (279.46M) uses 77.5% fewer trainable parameters. Relative to *V-1* (1.51B), it achieves an 81.5% reduction, and relative to V-2 (279.09M), it uses only 0.13% more trainable parameters. Despite this slight increase over *V-2*, *V-3* achieves performance improvements ranging from 0.12% to 1.95% across six evaluation metrics, five of which are statistically significant (Fig. 6). We have reported number of trainable parameters in the ablation study (Table 4).
> Conceptually, our method only updates the **small projection network and classifier**, while keeping the 1B-parameter LLM frozen; this yields **over an order-of-magnitude fewer trainable parameters** compared to full LLM fine-tuning.
>
> *Addition to the current revised version: (i) Improved paper clarity (including consistent bold notations, replacing vague phrase, etc.), (ii) experiments with additional backbone in the Appendix A.7 and Table 8, (iii) qualitative sample in Appendix A.3 Fig. 9*
>
> **Closing remark.**
> Thank you for the detailed and very constructive comments on clarity and evaluation. We have revised the paper accordingly and will incorporate the remaining clarifications in the final version. We hope these changes address your concerns and kindly ask you to consider revising your overall score if you now find the work clearer and more compelling.

---

### Official Review · Reviewer_ws6g · 2025-10-28

**Soundness:** 3
**Presentation:** 3
**Contribution:** 2
**Rating:** 4
**Confidence:** 5

**Summary:**

This paper proposes RAD3D-Prefix, a lightweight framework for radiology report generation from 3D CT scans. The method keeps a pretrained LLM frozen and learns a small anomaly-aware prefix projection module that maps 3D visual embeddings and multi-label abnormality logits into the LLM token space. The approach aims to reduce computational cost and overfitting while improving clinical alignment. Experiments on CT-RATE and INSPECT show competitive text-generation metrics and higher GREEN (clinical accuracy) scores compared to prior models such as CT2Rep and E3D-GPT.

**Strengths:**

The paper is clearly written and motivated by practical concerns in medical imaging, namely the high cost of fine-tuning large LLMs on limited 3D data. The idea of using prefix learning for frozen LLMs is simple and parameter-efficient, and the inclusion of anomaly logits provides some domain relevance. Experiments are extensive and cover both in-domain and out-of-domain datasets, with reasonable baseline comparisons.

**Weaknesses:**

- Despite solid engineering execution, the conceptual novelty is limited. Prefix-tuning and lightweight alignment for frozen LLMs are already well-established in both natural-image and medical domains. The proposed “anomaly-aware” extension is a concatenation of abnormality logits to the prefix embedding, which offers little methodological innovation or theoretical insight. The paper does not clearly articulate why this integration constitutes a new learning mechanism rather than a routine design choice.

- The empirical evidence is also weak in supporting strong claims. Reported gains in BLEU or ROUGE are small and often inconsistent across datasets. The primary metric improvement lies in GREEN, but the calculation and clinical validation of this metric are insufficiently described. Statistical significance, variance, and inter-observer consistency are missing, leaving uncertainty about the robustness of the results.

- The experimental scope is narrow. Both datasets are chest CT–based, so generalization to other modalities (MRI) or anatomical regions is untested. The model’s reliance on CT-CLIP and pretrained classifiers also confounds attribution of performance gains—improvements may stem from better visual encoders rather than the prefix design.

- The claimed anomaly awareness and interpretability are largely unsubstantiated. There is no qualitative analysis or visualization showing that the prefix module meaningfully encodes pathology patterns or improves factual correctness. The prefix mechanism acts as a generic adapter, not a clinically interpretable reasoning step.

**Questions:**

Please refer to the Weaknesses section.

**I am willing to raise my score according to the rebuttal.**

---

> ### Author Response · Authors · 2025-11-28
> **(1/3) Addressing Novelty, Empirical Evidence, Interpretability, and Modality Generalization**
>
> We thank the reviewer for the valuable feedback and thoughtful suggestions. Below, we provide a point-by-point response to each comment and summarize the corresponding revisions made to the manuscript.
>
> ***Q1/W1:** **Despite solid engineering execution, ... routine design choice.***
>
> **Response:** We agree that prefix-tuning and lightweight alignment for frozen LLMs are well established in natural-image and some medical settings. Our goal is not to propose a new theoretical paradigm, but to answer a domain-specific (a very important) question: *Can an anomaly-aware prefix, operating on 3D CT volumes, provide a lightweight yet clinically effective way to adapt LLMs that were never trained on radiology text?*  Please note that existing works often adopt frozen LLMs without investigating (i) whether medical data benefit from freezing, (ii) scaling behavior across model sizes, or (iii) the role of architectural choices such as anomaly conditioning. Our study directly addresses these gaps and introduces contributions that, to our knowledge, have not been explored in 3D medical imaging.
> Our contributions beyond prior work are:
> + We introduce a **multi-token, anomaly-aware prefix** where each token is constructed from *both* volumetric CT embeddings and **multi-label abnormality logits**, rather than visual features alone. A transformer layer over this sequence models cross-anomaly and image–logit interactions, enabling the prefix to separate normal and abnormal context.
> + We provide, to our knowledge, the **first comprehensive study of frozen vs fine-tuned LLMs in 3D CT report generation** across five LLM scales, showing that the behavior of frozen LLMs on medical 3D data deviates from what is reported in LLaVA/BLIP-2 on natural images.
>  + We demonstrate that this anomaly-aware prefix yields improvements in **GREEN, F1-RadGraph, and RaTEScore** over strong baselines using the same CT-CLIP encoder, attributing the gain specifically to the prefix design.
>
> We have further clarified in the introduction that our contribution is an empirically grounded, 3D medical instantiation of prefix learning rather than a new generic learning framework.
>
>  + *Systematic LLM-scale study in 3D radiology:* We conduct an exhaustive comparison across **five** LLMs from 96.1M to 1.6B parameters, with frozen vs fine-tuned setups, specifically on 3D CT, where pretraining data are entirely non-medical. This provides actionable guidance (fine-tune $<$ 1B, freeze 1B+) that has not been studied in 3D medical imaging.
>
> + Incorporating anomaly-aware prefix tokens yields improvement of 24.85%  in F1-RadGraph, 7.22% in RaTEScore and 33.2% in GREEN Score
> over R2GenGPT using a same-sized LLM and a standard visual-only mapping network.
>
> + Our Bootstrap analysis in the updated version (Sec. 4.5) demonstrates that the improvements in variant *V-3*  (with logits) over *V-2* (w/o logits) are statistically significant across five of the six evaluation metrics in both datasets. This confirms that the gains are consistent and unlikely to arise from sampling variability.
>
> + We report GREEN score which is explicitly designed for radiology reports and shown to correlate strongly with expert analysis. In the revised version, we have also included some GREEN Summary samples as well as a comparative analysis of matched findings between variants *V-2* and *V-3* (*Appendix A.4* and *Fig. 10*). Note that **RAD3D-Prefix** corresponds to the *V-3* variant in this comparison. The GREEN Score analysis across 3039 test cases shows that *V-3* produces 253 additional matched findings (1.43% improvement) compared to *V-2*, corresponding to approximately one additional finding every 12 cases. Given the sparsity and clinical importance of radiology findings, this improvement is clinically meaningful.
>
> In the camera-ready version, we will additionally derive an automatic error taxonomy (e.g., missing findings, incorrect location, incorrect negation) from GREEN outputs.
>
> *Addition to the current revised version: Bootstrap analysis in Sec. 4.5. GREEN Score summary samples and matched-findings analysis in Appendix A.4 and Fig. 10. We have revised the contribution summary to clearly articulate the core aspects of novelty.*

---

> ### Author Response · Authors · 2025-11-28
> **(2/3) Addressing Novelty, Empirical Evidence, Interpretability, and Modality Generalization**
>
> **Q2/W2: The empirical evidence is also weak ... robustness of the results.**
>
> **Response:** We appreciate the concern. Absolute gains in BLEU/ROUGE are indeed numerically modest, which is typical in radiology-report generation where even 0.5–1.0 point differences can correspond to noticeable clinical improvements.
>
> To assess robustness, we now include a **bootstrap analysis** with 5,000 resamples (*Sec. 4.5 and Fig. 6*), showing that RAD3D-Prefix achieves statistically significant improvements (non-overlapping 95% CIs) on **five out of six metrics** across both datasets.  This goes beyond point estimates and demonstrates that the gains are consistent rather than due to sampling noise.
>
> We also extended the GREEN description (*Appendix A.4*), summarizing its computation and citing its original source, where GREEN correlates strongly with expert assessment and has been adopted in recent radiology-report studies [1, 2]. Because all our scores (including GREEN) are computed automatically using the same pretrained models, **inter-observer variability** is not applicable} (no new human annotations are collected). However, we plan to add a small-scale radiologist study on a held-out subset in the camera ready version.
>
> *Addition to the revised version: (a) Bootstrap analysis (Sec. 4.5 and Fig. 6), (b) GREEN Score details (Appendix A.4 and Fig. 10)*
>
> [1] Nath, Vishwesh, Wenqi Li, Dong Yang, Andriy Myronenko, Mingxin Zheng, Yao Lu, Zhijian Liu et al. "Vila-m3: Enhancing vision-language models with medical expert knowledge." In Proceedings of the Computer Vision and Pattern Recognition Conference, pp. 14788-14798. 2025.
>
> [2] Liu, Kang, Zhuoqi Ma, Xiaolu Kang, Yunan Li, Kun Xie, Zhicheng Jiao, and Qiguang Miao. "Enhanced contrastive learning with multi-view longitudinal data for chest x-ray report generation." In Proceedings of the Computer Vision and Pattern Recognition Conference, pp. 10348-10359. 2025.
>
> **Q3/W3: The experimental scope is narrow .. than the prefix design.**
>
> **Response:** We agree that our current experiments are limited to chest CT, which reflects a deliberate focus: we wanted to target a **single, clinically important 3D modality** to precisely analyze the effect of anomaly-aware prefix learning and freezing/fine-tuning strategies for LLMs on previously unseen medical data.
>
> We emphasize that:
>   + The framework is **modular**: the CT-CLIP encoder and classifier can be replaced with any 3D encoder (for MRI, other anatomies) while keeping the same prefix mechanism.
>  + All baselines and variants share the **same CT-CLIP embeddings** and differ only in the mapping network and use of logits. Thus, improvements cannot be attributed to a better visual backbone but rather to the anomaly-aware prefix itself.
>  + Our LLM-based analysis, which examines freezing and fine-tuning strategies for smaller and larger LLMs, specifically targets medical datasets that pre-trained LLMs have not previously encountered. Since the core idea focuses on how LLMs pre-trained on natural image-text data can be optimized with medical image–text pairs, the framework is broadly applicable beyond chest CT scans.
> We will highlight extension to other modalities and anatomies as a concrete avenue for future work.

---

> ### Author Response · Authors · 2025-11-28
> **(3/3) Addressing Novelty, Empirical Evidence, Interpretability, and Modality Generalization**
>
> **Q4/W4: The claimed anomaly awareness ... interpretable reasoning step.**
>
> **Response:** Thank you for pointing this out. To support the effectiveness of our approach, we included UMAP visualizations comparing *V-2* (without logits) and *V-3* (with logits) variants (*Appendix A.8 and Fig. 12*). Additionally, we updated the manuscript with qualitative sample for all three variants (*Appendix Fig. 9*).
>
>  + To inspect the learned representation in the *V-2* and *V-3* variants, we projected the embeddings onto the three components and visualized them from multiple viewpoints in **UMAP**. Radiologic labels are heavily intermixed, as expected in a multi‑label setting, but show local enrichment along different parts of the manifold (e.g., bronchiectasis and interlobular septal thickening toward one extremity, pleural effusion and consolidation along central segments). Across *V-2* and *V-3* variants, the global topology is stable, however, *V-3*  yields a more compact S‑shaped ribbon with slightly reduced thickness, reflecting a more structured organization of the latent space while preserving the overall disease continuum. The UMAP analysis demonstrates that the anomaly-aware mechanism successfully creates unique vectors for structurally important diseases.
>  + We have included a **qualitative sample** in *Appendix A.3 Fig. 9*, comparing the three variants, *V-1*, *V-2*, and *V-3* and highlighting matched sentences where anomaly logits lead to more complete or more precise findings. In the camera-ready, we will include some more samples in the Appendix.
>   + By design, the prefix is a deterministic function of **multi-label abnormality logits** (e.g., effusion, consolidation, nodule). This makes the conditioning **anomaly-aware**: each token embeds both image context and explicit abnormality scores.
>   + Table 4 already shows that **removing these logits reduces GREEN and RadGraph performance**, demonstrating that the logits contribute to factual correctness.
>    + At analysis time, we can inspect which logits are active and ablate them to study how the generated reports change, providing a pragmatic interpretability handle (e.g., “what happens when the effusion logit is zeroed?”). In the camera-ready version, we will add qualitative examples illustrating how specific logits affect the mention of corresponding pathologies.
>
> *Addition to the current revised version: UMAP visualization in Appendix A.8 and Fig. 12. Qualitative sample comparing the three variants in Appendix A.3 Fig. 9*
>
> **Closing remark.**
> We appreciate your detailed technical feedback. We hope that the added statistical analysis, clearer novelty framing, qualitative  analysis and UMAP visualisations, and refined discussion of anomaly awareness address your concerns, and we kindly ask you to consider revising your score if you find these clarifications convincing.

---

### Official Review · Reviewer_NL8p · 2025-11-01

**Soundness:** 3
**Presentation:** 3
**Contribution:** 3
**Rating:** 4
**Confidence:** 2

**Summary:**

This paper addresses 3D CT report generation, identifying that fine-tuning large LLMs on small medical datasets leads to overfitting. To overcome this issue. The authors proposes RAD3D-Prefix as a solution. RAD3D-Prefix is a transformer-based prefix projector that fuses a 3D image embedding (from CT-CLIP) with multi-label abnormality logits and learned “prefix tokens” to a frozen LLM (LLaMA-3.2-1B) with the goal is to align volumetric CT features with textual report space while training only the projector, hence reducing overfitting on small medical datasets. Different variants of the architecture are proposed. Experiments on CT-RATE and INSPECT show that freezing larger LLMs with the proposed prefix improves report metrics compared to linear projection baselines and several prior systems (e.g., R2GenGPT, CT2Rep)

**Strengths:**

1. This is a well-executed paper. The core idea—to create an "anomaly-aware" prefix by concatenating visual features with multi-label classification logits is a simple but effective. The frozen encoder (CT-CLIP), lightweight projector, and frozen LLM are . The anomaly-aware logit fusion is simple and clinically motivated.
2. The paper has good structure and is relatively easy to follow.

**Weaknesses:**

1. Missing radiologist (human expert) assessment, no error taxonomy. Although several metrics have been used to evaluate the report quality, these N-gram metrics are known to be weak proxies for radiology quality. The authors are recommended to include a small evaluation set from domain expert to make sure that the proposed method actually work in the real-world setting.
2. The core idea (prefix learning) is known here, I feel the novelty here is mainly 3D + anomaly-logit fusion into a multi-token prefix with a frozen LLM. This is a useful contribution indeed, but largely incremental relative to linear-projector baselines so I recommend the authors to sharpen what is fundamentally new

**Questions:**

1. I wonder how do performance and compute trade off as you vary the learned prefix length and the projector’s capacity (layers/hidden size)? Could you show this.

I'd be happy to increase my score if the authors can address my concerns.

---

> ### Author Response · Authors · 2025-11-26
> **(1/2) Addressing Clinical Relevance, Novelty, and Performance Tradeoffs**
>
> We thank the reviewer for the insightful feedback and constructive suggestions. Below, we provide a detailed point-by-point response to each reviewer comment and summarize the updates implemented in the revised manuscript.
>
> **W1: Missing radiologist (human expert) ... work in the real-world setting.**
>
> **Response:** We fully agree that radiologist assessment is the gold standard, and that an explicit error taxonomy would further strengthen the clinical relevance. Unfortunately, organizing and running a new reader study (recruitment, grading) is not feasible within the rebuttal timeline (but we plan to add it in the camera ready). Following common practice in radiology report generation (CT-AGRG Di Piazza et al. 2024; E3D-GPT Lai et al. 2024), we rely on automatic but *clinically oriented* metrics.
>
> Beyond traditional n‑gram metrics, we report **GREEN** (0.1368↑), which was explicitly designed for radiology reports and shown to correlate strongly with expert ratings. In the current updated version, we further extended the evaluation with **F1-RadGraph** and **RaTEScore** (*Appendix Table 6 and Table 7*). Our method improves over a same-sized LLM with the R2GenGPT mapping network by **24.85%** in F1-RadGraph and **7.22%** in RaTEScore, and still yields **10.78%** and **1.02%** gains when we replace our transformer-based network with a simple linear layer using identical CT-CLIP embeddings and LLM. These gains target structural and factual correctness rather than only surface form.
>
> In the camera-ready version, we will additionally:
>
> + derive an **automatic error taxonomy** (e.g., missing findings, incorrect location, incorrect negation) from GREEN outputs. In the current revised version, we have already included some GREEN Summary samples as well as a comparative analysis of matched findings between variants *V-2* and *V-3* (*Appendix A.4 and Fig. 10*). Note that **RAD3D-Prefix** corresponds to the *V-3* variant in this comparison. Across 3039 test cases, *V-3* produces 253 additional matched findings (1.43% improvement) compared to *V-2*, corresponding to approximately one additional finding every 12 cases. Given the sparsity and clinical importance of radiology findings, this improvement is clinically meaningful.
> + outline our plan for a small-scale radiologist study on a held-out subset.
>
> We hope that, given the community’s reliance on GREEN/RadGraph-style metrics and the consistent improvements we report, the current evaluation is sufficient to judge the method’s practical value at this stage.
>
> Di Piazza, Theo, Carole Lazarus, Olivier Nempont, and Loic Boussel. "Ct-agrg: Automated abnormality-guided report generation from 3d chest ct volumes." In 2025 IEEE 22nd International Symposium on Biomedical Imaging (ISBI), pp. 01-05. IEEE, 2025.
>
> Haoran Lai, Zihang Jiang, Qingsong Yao, Rongsheng Wang, Zhiyang He, Xiaodong Tao, Wei Wei, Weifu Lv, and S Kevin Zhou. E3d-gpt: Enhanced 3d visual foundation for medical vision-language model. arXiv preprint arXiv:2410.14200, 2024.
>
>
> **W2: The core idea (prefix learning) is known here ... authors to sharpen what is fundamentally new.**
>
>  **Response:** We agree that prefix learning itself is not new and already state this in our contribution summary. Our work is novel in how this paradigm is instantiated and analyzed for 3D CT:
>
>    + **Anomaly-aware prefix for 3D CT:** We are, to our knowledge, the first to fuse *volumetric* CT embeddings and *multi-label abnormality logits* into a unified multi-token prefix, rather than using purely visual tokens. This explicitly exposes clinical concepts (e.g., effusion, consolidation) to the LLM.
>   + **Systematic LLM-scale study in 3D radiology:** We conduct an exhaustive comparison across **five** LLMs from 96.1M to 1.6B parameters, with frozen vs fine-tuned setups, specifically on 3D CT, where pretraining data are entirely non-medical. This provides actionable guidance (fine-tune < 1B, freeze 1B+) that has not been studied in 3D medical imaging.
>    + **Clinically grounded improvements under a lightweight projector:** Using the same CT-CLIP encoder for all methods, our lightweight mapping network consistently improves GREEN, F1-RadGraph, and RaTEScore over strong baselines, demonstrating that the gains come from the anomaly-aware prefix rather than a heavier backbone.
>
> *We have sharpen the contribution summary to clearly articulate these three aspects as the core novelty.*

---

> ### Author Response · Authors · 2025-11-26
> **(2/2) Addressing Clinical Relevance, Novelty, and Performance Tradeoffs**
>
> **Q1: I wonder how do performance and compute trade off ... Could you show this.**
>
> **Response:** We thank the reviewer for the suggestion. We have conducted an ablation study with prefix lengths 2,5,7, 10, and 14 and 2, 4, 8, and 16 layers, analyzing tradeoffs between performance and computational overhead. The results are reported in the *Appendix A.6 and Fig. 11*.
>
> *Prefix Length:* From Fig. 11 (a), it can be observed that increasing prefix length significantly improves performance up to a threshold, after which gains saturate. The most significant improvement occurs between prefix lengths 2 and 5 with gains above 60% across five metrics. Beyond prefix length 5, the improvements are marginal despite increased parameter count.
>
> *Number of Layers:* Fig. 11 (b) shows that while increasing the number of layers dramatically increases trainable parameters, there is no measurable performance gain across NLG metrics. Although a slight improvement is observed in the GREEN metric, the gain is not proportional to the increase in parameters.
>
> *Addition to the revised version: Appendix A.6 (Analysis of Prefix Length and Layers Vs. Performance and Trainable Parameters) and Fig.11.*
>
> **Closing remark:** We sincerely thank you for the constructive feedback. We hope that the strengthened clinically grounded evaluation and clarified novelty address your main concerns, and we would be very grateful if you might consider updating your overall score in light of these clarifications.

---

### Author Response · Authors · 2025-12-04
**Consolidated Summary of Responses and Revisions for the Area Chair**

We thank the reviewers for their constructive feedback and thoughtful suggestions. We have addressed the raised concerns by incorporating the suggestions, clarifying ambiguities, and adding further supporting results. Below is a summary of our responses and revisions.

- **Clarification of Novelty (Addressing Reviewers NL8p, bop9, ws6g)**
   - While prefix-tuning is established, our approach is the **first to introduce a multi-label, anomaly-aware prefix** specifically designed for 3D CT.  It explicitly exposes clinical concepts (e.g., effusion, consolidation) to the LLM.
    - Our work provides the **first systematic frozen vs. fine-tuned LLM scaling study** (96.1M–1.6B) for 3D medical imaging, producing actionable guidance: *fine-tune $<$1B, freeze 1B+*, **contrasting with findings in natural image domains**.
     - Using the same CT-CLIP encoder for all methods, our lightweight mapping network consistently outperforms strong baselines, demonstrating that **the gains come from the anomaly-aware prefix rather than a heavier backbone**.
        - *We have sharpen the contribution summary (Page-3) to clearly articulate these three aspects as the core novelty.*

- **Robustness of Empirical Results (Addressing Reviewers NL8p, bop9, ws6g)**
   - To address concerns about *"marginal gains", "weak empirical evidence" and "anomaly-aware part was just a generic adapter"*, we added a **Bootstrap Analysis (5,000 iterations)**. Across both datasets, variant *V-3* (used in RAD3D-Prefix) yields statistically significant gains on 5/6 metrics over *V-2*, highlighting the importance of incorporating classification logits.
    - We **included two additional metics, F1-RadGraph and RaTEScore**, specialized to evaluate clinical efficacy. Our model reports improvements in F1-RadGraph (+24.85\%), RaTEScore (+7.22\% ), GREEN (+33.2\%) over R2GenGPT (with same LLM size).
     - On 3039 cases, GREEN score summary reveals that anomaly-aware variant (*V-3*) **adds 253 extra matched findings (1 every 12 cases)** over *V-2*.
     - While a full human evaluation was not feasible within the rebuttal window, we will add a small-scale expert review and an automated error taxonomy in the camera-ready version.
        - *We added Bootstrap analysis (Sec. 4.5 and Fig. 6), GREEN summary sample (Appendix Fig. 10), matched-findings count comparing V-2 (w/o logits) and V-3 (w/ logits) (Appendix A.4) and results on additional metrics (Appendix Table 6 and Table 7).*

- **Interpretability \& Visual Evidence (Addressing Reviewers ws6g , BUQA)**
   - To address *"prefix mechanism acts as a generic adapter, not a clinically interpretable reasoning step"*, we **added UMAP visualizations** showing clear geometric differences with and without logits, thus visually validating that the prefix encodes meaningful clinical semantics.
    - We added **qualitative comparisons (*V-1* vs *V-2* vs *V-3*)** showing more complete and precise clinical findings.
       - *We added UMAP visualizations (Appendix A.8 and Fig. 12) and qualitative comparisons (Appendix A.3 and Fig. 9).*

- **Ablations \& Efficiency (Addressing Reviewers BUQA, bop9)**
  - **Prefix length and projector depth ablations:** performance improvements are marginal after prefix length 5; deeper networks add parameters with negligible gain.
  - Updated Table 4 with trainable parameters: **RAD3D-Prefix (*V-3*) uses 77-82% fewer trainable parameters** than fine-tuning the LLM.
   - **Added another backbone results**, specifically an 8B LLM (DeepSeek-R1-Distill-LLaMA-8B ), showing *scalability and generality*.
     - *We added prefix length and projector depth ablations (Appendix A.6 and Fig. 11), updated Table 4 with trainable parameters, included results for another backbone DeepSeek-R1-Distill-LLaMA-8B (Appendix A.7 and Table 8).*

- **Clarification on Out-of-Domain (OOD) Performance (Addressing Reviewer bop9)**
   - Although scores on INSPECT are lower due to domain shift, RAD3D-Prefix still provides:
+8.16\% GREEN, +75.7\% METEOR, +19.9\% ROUGE over CT2Rep (3D SoTA).
   - Bootstrap confirms improvements are **statistically reliable**.
     - *Included Bootstrap analysis with forest plots (Sec. 4.5 and Fig. 6).*

- **Writing, Organization \& Presentation (Addressing Reviewer BUQA)**
   - Revised the manuscript **to fix inconsistent notation** (bolding vectors) and replaced vague terminology ("bridge the gap") with precise technical definitions regarding token space projection.
    - Provided explicit mathematical shapes, definitions, and data flows: *How visual + logit fusion works on unseen anomalies, What “structured sequence” means, How target embeddings $\hat{R}$ are obtained, How gradients flow through the frozen LLM.*

---

### Meta-Review · Area_Chair_gFvf · 2025-12-22

**Summary:**

All four reviewers expressed predominantly negative opinions. NL8p, ws6g, and bop9 raised concerns regarding the limited contribution of the work. The proposed anomaly‑aware prefix approach has already been explored extensively in prior studies, such as promptMRG (AAAI 2024), yet the manuscript did not provide a sufficient literature review or comparative analysis against these existing methods. All reviewers pointed out issues in the experimental design from different aspects, including missing radiologist assessment (NL8p), weak empirical evidence (ws6g), narrow experimental scope (ws6g), limited and unclear evaluations (BUQA), and poor generalization on out‑of‑domain data (bop9). In addition, the AC notes that the main claim of the paper is to introduce a lightweight anomaly‑aware prefix projection to align image and text features. However, the classification logits may directly leak information to the LLM rather than fundamentally resolving the alignment problem. Furthermore, there is no clear evidence that this method achieves actual alignment between the LLM and the vision space, and this would require additional experimental validation and deeper discussion. Given these major concerns, I believe the manuscript is not yet suitable for acceptance and therefore recommend rejection.

**Reviewer Concerns:**

The authors have provided detailed responses to the reviewers’ comments, and many issues have been at least partially resolved, such as the additional clarifications regarding the methodology and experimental design. However, concerns about the novelty of the work remain. At present, the manuscript and the response still do not sufficiently support the contribution of the core method. Further improvement is needed in the literature review and comparative analysis against related work.

**Reviewer Scores:**

Considering that key concerns remain unresolved, I think the reviewers are unlikely to have a clear motivation to change their scores.

---

### Decision · Program_Chairs · 2026-01-26

Reject